# HILBERT: RECURSIVELY BUILDING FORMAL PROOFS WITH INFORMAL REASONING

**Sumanth Varambally**[*]    **Thomas Voice**    **Yanchao Sun**    **Zhifeng Chen**    **Rose Yu**[†]
UC San Diego          Apple           Apple           Apple           UC San Diego

**Ke Ye**[†]
Apple

## ABSTRACT

Large Language Models (LLMs) demonstrate impressive mathematical reasoning abilities, but their solutions frequently contain errors that cannot be automatically checked. Formal theorem proving systems such as Lean 4 offer automated verification with complete accuracy, motivating recent efforts to build specialized prover LLMs that generate verifiable proofs in formal languages. However, a significant gap remains: current prover LLMs solve substantially fewer problems than general-purpose LLMs operating in natural language. We introduce HILBERT, an agentic framework that bridges this gap by combining the complementary strengths of informal reasoning and formal verification. Our system orchestrates four components: an informal LLM that excels at mathematical reasoning, a specialized prover LLM optimized for Lean 4 tactics, a formal verifier, and a semantic theorem retriever. Given a problem that the prover is unable to solve, HILBERT employs recursive decomposition to split the problem into subgoals that it solves with the prover or reasoner LLM. It leverages verifier feedback to refine incorrect proofs as necessary. Experimental results demonstrate that HILBERT substantially outperforms existing approaches on key benchmarks, achieving 99.2% on miniF2F, 6.6% points above the best publicly available method. HILBERT achieves the **strongest known result** from a publicly available model on PutnamBench. It solves 462/660 problems (70.0%), outperforming proprietary approaches like SeedProver (50.4%) and achieving a 422% improvement over the best publicly available baseline. Thus, HILBERT effectively narrows the gap between informal reasoning and formal proof generation. Code is available at `https://github.com/Rose-STL-Lab/ml-hilbert`.

## 1 INTRODUCTION

General-purpose Large Language Models (LLMs) have achieved dramatic improvements in mathematical understanding. Reasoning LLMs like GPT-5 and Gemini 2.5 Pro attain near-perfect performance on high-school olympiad exams such as AIME and can solve a significant proportion of competitive undergraduate-level problems from the Putnam exam (Dekoninck et al., 2025). These systems also show promise on research-level benchmarks like FrontierMath (Glazer et al., 2024; OpenAI, 2025).

However, several fundamental limitations severely constrain their practical utility. These systems frequently hallucinate, producing confident-sounding but ultimately incorrect solutions. Even when the final answers are correct, the underlying reasoning often contains serious flaws: "proving" by example, logical fallacies, unjustified assumptions, and calculation errors (Petrov et al., 2025; Guo et al., 2025; Mahdavi et al., 2025; Balunović et al., 2025). Manual verification of generated proofs is time-consuming, difficult, and error-prone. Although recent advances show LLM-based verifiers can approach human-level performance (Guo et al., 2025; Dekoninck et al., 2025), they remain fallible due to hallucinations and silent failures (Mahdavi et al., 2025; Petrov et al., 2025).

---

[*]Work done during internship at Apple. Correspondence: svarambally@ucsd.edu, roseyu@ucsd.edu
[†]Equal co-supervisors.

Figure 1: **The HILBERT algorithm.** Given a target theorem, HILBERT attempts formal proof generation with the prover. Upon failure, it decomposes the problem into subgoals and tries to solve them with the prover, followed by the reasoner (shallow solve). If both strategies fail, it resorts to recursive decomposition until all subgoals are resolved.

Formal theorem proving systems such as Lean 4 (Moura & Ullrich, 2021) offer a promising solution, as they can automatically and unambiguously verify whether a proof is correct when expressed in a formal language (Kontorovich, 2025). This capability has spurred the development of purpose-built prover LLMs (Polu & Sutskever, 2020), with substantial research focused on developing specialized models for generating formal Lean 4 proofs (Yang et al., 2023; Xin et al., 2024a;b; 2025a; Ren et al., 2025; Dong & Ma, 2025; Wang et al., 2025). The best open prover models achieve over 90% pass rate on miniF2F (Zheng et al., 2021) and solve 86 of 657 problems on the challenging PutnamBench (Tsoukalas et al., 2024). Proprietary systems such as AlphaProof (AlphaProof & AlphaGeometry, 2024) and SeedProver (Chen et al., 2025) demonstrate this paradigm's potential, achieving a silver-medal performance on problems from the International Mathematical Olympiad (IMO).

Despite this progress, a significant performance gap remains between specialized prover LLMs and general-purpose reasoning LLMs. For example, Dekoninck et al. (2025) found through human verification that reasoning LLMs can solve approximately 83% of PutnamBench problems informally, while the best publicly available prover LLMs achieve only 13% with formal proofs. General-purpose LLMs excel at informal mathematical reasoning and understand formal language syntax well enough to write effective proof sketches and short proofs (Ren et al., 2025; Liang et al., 2025). However, they struggle with full formal program synthesis, achieving only 49.1% pass rate (with 16384 attempts) on miniF2F (Zhou et al., 2025b). Conversely, specialized prover LLMs excel at producing syntactically correct formal proofs for standalone theorems, but are brittle at language-intensive tasks like leveraging existing theorems or error correction (Liang et al., 2025).

To address this gap, several works have explored incorporating informal reasoning from general-purpose LLMs to augment formal theorem-proving capabilities. Early approaches like DSP (Jiang et al., 2022) and LEGO-Prover (Wang et al., 2023) used general-purpose LLMs to propose proof sketches, with automated theorem provers (ATPs) filling formal components, but were limited by heuristics-based ATP capabilities. DSP+ (Cao et al., 2025) extended this approach using modern prover LLMs for intermediate steps. However, these methods struggle with complex subgoals due to shallow, single-layer decomposition. They break down the original problem but cannot further decompose subgoals that remain too difficult to solve directly. Recent agentic frameworks including COPRA (Thakur et al., 2024), Prover-Agent (Baba et al., 2025), and ProofCompass (Wischermann et al., 2025) iteratively construct proofs using informal reasoning with feedback from the formal verifier. Although these methods show promise, their performance still significantly lags behind general-purpose reasoning LLMs.

We introduce HILBERT, an agentic framework that bridges informal reasoning with formal verification (Figure 1). It orchestrates four key components: a general-purpose reasoning LLM, a prover LLM, a verifier, and a semantic theorem retriever. Given a mathematical problem, HILBERT first retrieves relevant theorems from Mathlib (mathlib Community, 2020) and generates a detailed informal proof using the reasoner. It then creates a Lean 4 proof sketch decomposing the problem into manageable subgoals. For each subgoal, HILBERT employs a two-stage approach: attempting formal proof generation with the prover, then falling back to the reasoner augmented with retrieved theorems. When both stages fail, the system recursively decomposes problematic subgoals into smaller problems. Throughout the process, HILBERT leverages the reasoner to interpret compilation errors, suggest corrections, and guide proof refinement at inference-time. We summarize our main contributions below.

- We design HILBERT, a multi-turn agentic framework that systematically combines informal mathematical reasoning with formal proof verification, closing the performance gap between these two paradigms.

- We conduct comprehensive experiments on MiniF2F and PutnamBench, achieving state-of-the-art performance on both benchmarks. HILBERT reaches 99.2% pass rate on miniF2F (6.6 points above the best public method) and solves 462/660 PutnamBench problems (70.0%), outperforming proprietary systems like SeedProver (50.4%) and achieving over $4\times$ improvement versus the best open-source baseline.

- Through extensive ablation studies, we validate the effectiveness of our key technical contributions: the recursive decomposition procedure for breaking down complex proofs and the retrieval-augmented generation mechanism for enhanced reasoning capabilities.

## 2 RELATED WORK

Automated Theorem Provers (ATPs) are computational systems designed to automatically discover proofs of mathematical theorems. Traditional approaches have primarily relied on symbolic reasoning methods (Robinson, 1965; McCune, 2003; Schulz, 2002) and integration tools like Sledgehammer that connect ATPs with interactive proof assistants (Blanchette et al., 2013; Czajka & Kaliszyk, 2018). Recently, LLMs have emerged as a promising new tool for automated theorem proving (Polu & Sutskever, 2020; Yang et al., 2024).

**Prover LLMs.** The general principle is to train specialized prover LLMs on large datasets of formal proofs, most prominently for the Lean (Moura & Ullrich, 2021) theorem prover. Some prominent models include GPT-f (Polu & Sutskever, 2020), ReProver (Yang et al., 2023), DeepSeek Prover family of models (Xin et al., 2024a;b; Ren et al., 2025), ABEL (Gloeckle et al., 2024), Goedel Prover V1 and V2 (Lin et al., 2025a;b), BFS Prover V1 and V2 (Xin et al., 2025a;b), STP-Prover (Dong & Ma, 2025) and Kimina Prover (Wang et al., 2025). These models are trained by curating a substantial corpus of formal proofs and performing some combination of supervised finetuning and reinforcement learning. Several approaches have enhanced these models by incorporating subgoal decomposition into the training process (Zhao et al., 2023; 2024; Ren et al., 2025), while POETRY (Wang et al., 2024) and ProD-RL (Dong et al., 2024) employ recursive problem decomposition. Proprietary prover LLMs like AlphaProof (AlphaProof & AlphaGeometry, 2024) and SeedProver (Chen et al., 2025) have pushed the frontier further, achieving a silver-medal performance on problems from the International Mathematics Olympiad (IMO). Still, significant performance gaps remain between specialized prover models and general-purpose LLMs in mathematical reasoning capabilities (Dekoninck et al., 2025).

**Using Informal LLMs for Formal Theorem Proving.** Several previous works have attempted to incorporate informal reasoning from general-purpose LLMs to improve formal reasoning abilities. DSP (Jiang et al., 2022) used the Codex LLM to propose proof sketches in Isabelle, with intermediate steps filled in by Sledgehammer. LEGO-Prover (Wang et al., 2023) extended this framework to handle a growing skill library of intermediate theorems for retrieval-augmented proving. Liang et al. (2025) argue that general purpose reasoning LLMs are more effective at decomposing problems into simpler subgoals compared to prover LLMs. Our work extends upon this observation by using informal reasoners to recursively build proof sketches to break the problem down into simpler sub-problems that can be handled by a prover or reasoning LLM.

Several works have also proposed using an informal LLM in an agentic framework for automated theorem proving. COPRA (Thakur et al., 2024) queries an informal LLM to construct proofs tactic by tactic, incorporating execution feedback, search history, and retrieved lemmas into subsequent prompts. Prover-Agent (Baba et al., 2025) uses a small informal reasoning model to produce proof steps and lemmas, which are autoformalized and solved using a prover LLM. Feedback from Lean is used to iteratively refine incorrect proofs. ProofCompass (Wischermann et al., 2025) enhances prover LLMs by adding informal proof steps as comments in the input. When proof attempts fail, it analyzes these failures to extract intermediate lemmas that enable effective problem decomposition. DeltaProver (Zhou et al., 2025b) introduces a custom Domain-Specific Language to perform subgoal decomposition, and iteratively repair the generated proof using verifier feedback. Notably, it only uses an informal LLM and does not rely on prover LLMs. In contrast, our work demonstrates that prover

LLMs become highly effective tools when orchestrated in an appropriately designed multi-agent framework.

## 3  HILBERT SYSTEM

In this section, we detail HILBERT, a multi-agent system that bridges informal mathematical reasoning and formal verification by orchestrating general-purpose reasoning LLMs with specialized prover LLMs. Our approach uses recursive subgoal decomposition to break complex theorems into simpler subgoals that can be proven and combined, achieving performance exceeding either approach in isolation.

### 3.1  COMPONENTS

Before we describe the inference algorithm, we first describe the components that HILBERT orchestrates.

**Reasoner.** A general-purpose reasoning LLM to write informal proofs, proof sketches in Lean, and in certain instances, a formal proof. In our work, we use Google Gemini 2.5 Flash and Pro (Comanici et al., 2025) due to their superior mathematical reasoning capabilities (Zhou et al., 2025b; Dekoninck et al., 2025). We also experiment with the open-source `gpt-oss-120b` model (Agarwal et al., 2025) to assess the generalizability of our approach beyond proprietary models.

**Prover.** A specialized prover LLM to write formal proofs given a formal theorem statement. In our work, we use DeepSeek-V2-7B (Ren et al., 2025) and Goedel-Prover-V2 32B (Lin et al., 2025b).

**Verifier.** A formal language verifier to check the correctness of the theorem statements and proofs. We use the Kimina Lean Server (Santos et al., 2025) with Lean v4.15.0 and Mathlib v4.15.0.

**Retriever.** A semantic search engine to retrieve relevant theorems from Mathlib (mathlib Community, 2020) built using sentence transformers (`all-mpnet-base-v2` (Song et al., 2020)) and FAISS (Douze et al., 2024) indexing. The system computes cosine similarity between query embeddings and pre-computed embeddings of informal theorem descriptions from the `mathlib_informal` (Gao et al., 2024) dataset, providing a simple yet effective alternative to custom retrieval models (Gao et al., 2024; Lu et al., 2025).

### 3.2  ALGORITHM

Given a formal statement in Lean 4, we first attempt direct proof using the Prover. It generates $K_{\text{initial proof}} = 4$ candidate proofs, which we verify using the Verifier. If any proof is valid, we return it immediately. When direct proof attempts fail, we use the Reasoner to decompose the problem into simpler subproblems and assemble them into a valid proof strategy. Figure 2 provides an overview of this stage.

#### 3.2.1  SUBGOAL DECOMPOSITION

**Step 1** (Theorem Retrieval). Given the formal statement, we prompt the Reasoner to produce $s = 5$ search queries to look for theorems that might help simplify the proof strategy. For each search query, we use the Retriever to retrieve the top $m = 5$ most semantically similar theorems and tactics from Mathlib. We again query the Reasoner to select only the relevant theorems from the fetched search results.

**Step 2** (Formal Proof Sketch Generation). We prompt the Reasoner to produce a detailed informal proof using the retrieved theorems. With this proof supplied in-context, we ask the Reasoner to generate a Lean 4 proof sketch that decomposes the problem into simpler subproblems represented as `have` statements. All subgoals are initially filled with `sorry`, a placeholder keyword that Lean can temporarily treat as a proof of the subgoal. We verify that the proof sketch is valid using the Verifier and leverage its feedback to correct any errors. We generate a maximum of $K_{\text{sketch attempts}} = 4$ sketch attempts for each input theorem.

**Step 3** (Subgoal Extraction). The Reasoner extracts subgoals from the proof sketch, converting them into independent theorem statements with relevant context from the original problem and preceding

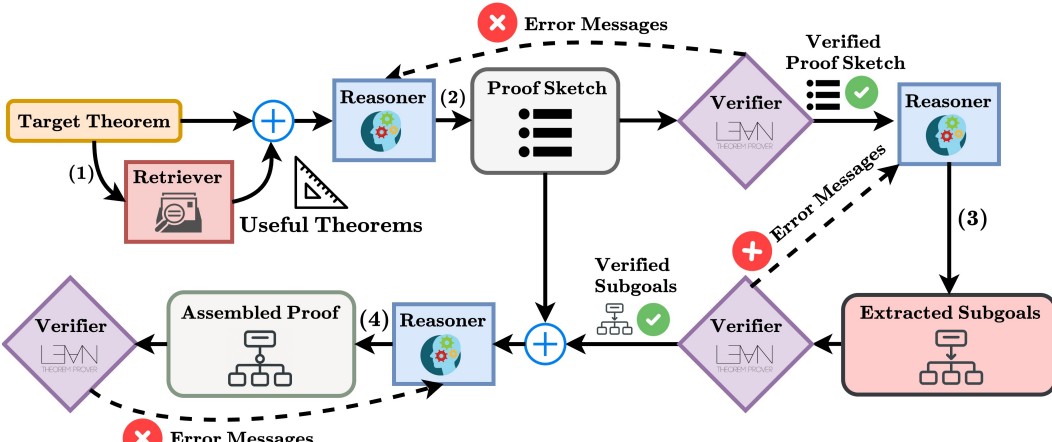

Figure 2: **Subgoal Decomposition:** Given a theorem statement, HILBERT: (1) retrieves relevant theorems from Mathlib using semantic search, (2) generates a formal proof sketch with subgoals marked as `have` statements with `sorry` placeholders, (3) extracts these subgoals as independent theorem statements, and (4) assembles the proof by replacing `sorry` placeholders with calls to the subgoal theorems. Verifiers ensure correctness at each stage. The error correction loops are indicated by dotted lines.

subgoals. As before, we use `sorry` for the proof. We verify completeness by counting `have` statements in the proof sketch and ensuring that all of them are extracted. In case any of them are missing, we prompt the Reasoner to extract the missing subgoals. Each extracted theorem undergoes syntax verification using the Verifier. When errors occur, we provide error messages in-context to the Reasoner for correction. This approach proves more reliable than parsing source code directly or extracting subgoals from Lean 4's proof state data structure (InfoTree) (Liang et al., 2025). It also enables us to prune irrelevant premises from the theorem statement of the subgoal.

**Step 4** (Sketch Assembly from Subgoals). We provide the Reasoner with the extracted subgoal theorem statements (which contain `sorry` placeholders) and validated proof sketch. The Reasoner produces an assembled proof for the target theorem by replacing each `sorry` placeholder in the proof sketch with calls to the corresponding subgoal theorem. We then verify both the subgoal theorem statements and the assembled proof together using the Verifier to ensure the overall structure is sound. We check for errors using the Verifier and correct them through iterative feedback with the Reasoner. This guarantees that after all subgoals are proven, we will have a complete proof of the given theorem.

### 3.2.2 SUBGOAL VERIFICATION

At this stage, we have a valid theorem proof structure and a list of subgoals that, if proven, complete the original proof. However, the mathematical correctness and provability of these subgoals remain unverified. For each subgoal, we execute the following verification and proof process:

**Step 1** (Prover Attempts). We first attempt to prove each subgoal directly using the Prover, generating $K_{\text{formal proof}} = 4$ candidate proofs and verifying them with the Verifier. If any generated proof is valid, we accept it and proceed to the next subgoal.

**Step 2** (Correctness Verification). For subgoals that cannot be directly proven, we prompt the Reasoner to evaluate whether the subgoal is mathematically correct and whether the formal statement is formulated correctly and is provable. If the Reasoner identifies the subgoal as mathematically incorrect, unprovable, or poorly formulated, we flag it for correction and return to refine the original proof sketch, repeating all steps from Section 3.2.1 onwards with the identified issues incorporated as feedback. Apart from mathematical errors, some common failure modes detected by the Reasoner at this stage include missing hypotheses or conditions in the subgoal theorem statement, and atypical behavior due to the Lean type system, such as truncation of natural numbers[1].

---

[1] `https://lean-lang.org/doc/reference/latest/Basic-Types/Natural-Numbers/`

We prioritize direct Prover attempts over Reasoner verification because the Prover models are computationally cheaper, and a valid proof automatically confirms mathematical correctness. Empirically, we observe that a significant proportion of generated subgoals can be successfully proven by the Prover. Step 1 ensures that we save on the computational costs of the expensive Reasoner model for verification on the successful subgoals.

**Step 3** (Shallow Solve). After Step 1 fails and Step 2 confirms subgoal correctness, we employ a Reasoner model for a "shallow solve" approach that writes short proofs for subgoals the Prover could not directly solve. We retrieve relevant theorems from the Mathlib library and ask the Reasoner to write a formal proof for the subgoal. The Reasoner iteratively refines proofs based on Verifier feedback for up to $K_{\text{proof correction}} = 6$ passes. When compilation errors indicate missing or incorrect theorem references, we retrieve additional relevant theorems. To preserve computational resources, we terminate this step if an incorrect proof exceeds the length threshold $K_{\text{max shallow solve length}} = 30$ lines, as excessively long proofs indicate the need for further decomposition. This entire shallow solve process repeats for up to $K_{\text{informal passes}} = 6$ attempts until we obtain a successful proof or exhaust all attempts.

**Step 4** (Recursive Decomposition and Proof Assembly). If subgoals remain unproven after Steps 1-3, we recursively apply the subgoal decomposition process (Section 3.2.1) to break them down further. Each subgoal is subdivided until it is either successfully proven or we reach the maximum recursion depth $D$. Should all subgoals become proven, we proceed to create a complete proof for the given theorem by stitching together the proofs for all subgoals and the assembled proof outline from Step 4 of subgoal decomposition. This is done by concatenating the proofs of the subgoals with the assembled proof produced in Step 4 of subgoal decomposition (Section 3.2.1). Any remaining unsolved subgoals at this point trigger a failed proof attempt, prompting us to restart the subgoal decomposition process for the theorem.

The complete algorithm is presented in Algorithm 1. For implementation details, particularly parallelization strategies, refer to Section A.3.

# 4 EXPERIMENTAL RESULTS

## 4.1 MAIN RESULTS

**MiniF2F.** The MiniF2F dataset (Zheng et al., 2021) is a 488 problem dataset comprising of high-school mathematics competition problems. Some problems are particularly challenging, sourced from the AMC, AIME and IMO competitions. We benchmark on the 244 problems from the test split of MiniF2F. We use recursion depth $D = 5$ for all our experiments. For the Prover, we instantiate HILBERT with two LLMs: DeepSeek-Prover-V2-7B (Ren et al., 2025), representing a relatively weaker model, and Goedel-Prover-V2-32B (Lin et al., 2025b), representing a stronger one. This pairing allows us to compare performance across different capability levels. For the Reasoner, we analogously employ Google's Gemini 2.5 Flash, Gemini 2.5 Pro (Comanici et al., 2025) and OpenAI's `gpt-oss-120b`. The results are presented in Table 1.

HILBERT demonstrates strong performance across all model configurations. Our top-performing setup combines Gemini 2.5 Pro with Goedel-Prover-V2-32B, achieving a 99.2% pass rate and failing on only two problems (AMC 12A 2020 Problem 25 and IMO Shortlist 2007 Problem A6). Even with weaker formal provers, HILBERT maintains impressive results: pairing DeepSeek-Prover-V2-7B with Gemini 2.5 Pro yields 98.4%, while using Gemini 2.5 Flash achieves 96.7%. HILBERT also generalizes to open-source informal reasoners: using `gpt-oss-120b` as the informal reasoner paired with Goedel-Prover-V2-32B achieves 90.8%, a 16.2% improvement over the pass@4 baseline. Notably, the choice of informal reasoner appears more critical than prover strength. Gemini 2.5 Pro consistently outperforms Flash variants by 3-4%, a larger gap than observed between different prover models, and both substantially outperform the open-source `gpt-oss-120b` variant. Compared to standalone base provers at pass@4, our approach delivers substantial improvements ranging from 16.2% to 37.1%.

**PutnamBench.** PutnamBench is a challenging theorem-proving benchmark comprising 660 problems from the William Lowell Putnam Mathematical Competition from 1962 to 2024. It contains undergraduate-level problems across Algebra, Analysis, Number Theory, Geometry, Linear Algebra,

| Method | Pass Rate |
|---|---|
| STP (Dong & Ma, 2025) (pass@3200) | $65.0\% \pm 0.5\%$ |
| (pass@25600) | 67.6% |
| Kimina-Prover-8B (Wang et al., 2025) (pass@32) | 78.3% |
| Kimina-Prover-72B (pass@1024) | 87.7% |
| w/ TTRL | 92.2% |
| Gemini 2.5 Pro (pass@16384) | 49.1% |
| Delta Prover (Zhou et al., 2025b) (pass@16384) | 95.9% |
| Seed Prover (Chen et al., 2025) | 99.6% |
| BFS-Prover-V2-32B w/Planner (Xin et al., 2025b) | 95.1% |
| Goedel-Prover-SFT (Lin et al., 2025a) (pass@3200) | 62.7% |
| Goedel-Prover-V2-8B (Lin et al., 2025b) (pass@8192) | 90.2% |
| w/ self-correction (pass@1024) | 89.3% |
| Goedel-Prover-V2-32B (pass@4) | $74.6\% \pm 1.2\%$ |
| (pass@8192) | 92.2% |
| w/ self-correction (pass@1024) | 92.6% |
| HILBERT (gpt-oss-120b) + Goedel-Prover-V2-32B | 90.8% [+16.2%] |
| HILBERT (Gemini 2.5 Flash) + Goedel-Prover-V2-32B | 94.7% [+20.1%] |
| HILBERT (Gemini 2.5 Pro) + Goedel-Prover-V2-32B | 99.2% [+24.6%] |
| DeepSeek-Prover-V2-7B (CoT) (Ren et al., 2025) (pass@8192) | 82.0% |
| DeepSeek-Prover-V2-7B (non CoT) (pass@4) | $61.3\% \pm 0.2\%$ |
| (pass@8192) | 75.0% |
| DeepSeek-Prover-V2-671B (pass@8192) | 88.9% |
| HILBERT (Gemini 2.5 Flash) + DS Prover-V2-7B (non-CoT) | 96.7% [+35.4%] |
| HILBERT (Gemini 2.5 Pro) + DS Prover-V2-7B (non-CoT) | 98.4% [+37.1%] |

Table 1: **Results on the MiniF2F-Test dataset.** Improvements shown in brackets for HILBERT are calculated relative to the pass@4 baseline for each prover family. Note: Delta Prover and Seed Prover are proprietary methods and not publicly available to use. Gemini 2.5 Pro result obtained from Zhou et al. (2025b)

| Model | # Solved Problems | % Solved Problems |
|---|---|---|
| Goedel-Prover-SFT (Lin et al., 2025a) (pass@512) | 7/644 | 1.1% |
| ABEL (Gloeckle et al., 2024) (pass@596) | 7/644 | 1.1% |
| Self-play Theorem Prover (Dong & Ma, 2025) (pass@3200) | 8/644 | 1.2% |
| Kimina-Prover-7B-Distill (Wang et al., 2025) (pass@192) | 10/657 | 1.5% |
| DSP+ (Cao et al., 2025) (pass@128) | 23/644 | 3.6% |
| Bourbaki (Zimmer et al., 2025) (pass@512) | 26/658 | 4.0% |
| DeepSeek-Prover-V2 671B (Ren et al., 2025) (pass@1024) | 47/657 | 7.1% |
| SeedProver (Chen et al., 2025) | 331/657 | 50.4% |
| Goedel-Prover-V2-32B (self-correction) (Lin et al., 2025b) (pass@184) | 86/644 | 13.4% |
| HILBERT (gpt-oss-120b) + Goedel-Prover-V2-32B | 88/660 | 13.3% |
| HILBERT (Gemini 2.5 Pro) + Goedel-Prover-V2-32B | 462/660 | 70.0% |

Table 2: **Results on the PutnamBench dataset.** We benchmark on the September 2025 version containing 660 problems.

Combinatorics, Abstract Algebra, Probability, and Set Theory. Given the high computational cost of evaluating on this dataset, we only experiment with the strongest configuration of HILBERT, (HILBERT with Gemini 2.5 Pro and Goedel-Prover-V2-32B). As before, we set $D = 5$. Our results are presented in Table 2.

| Method | Retrieval | Pass Rate | # Reasoner Calls | # Prover Calls | # Reasoner Tokens | # Prover Tokens |
|---|---|---|---|---|---|---|
| HILBERT+ DeepSeek-Prover-V2-7B | ✓ | **98.4%** | **420** | **205** | **1.9M** | **0.3M** |
| HILBERT+ DeepSeek-Prover-V2-7B | ✗ | 97.1% | 426 | 290 | 2.1M | 0.4M |
| HILBERT+ Goedel-Prover-V2-32B | ✓ | **99.2%** | **548** | **391** | **2.3M** | 1.3M |
| HILBERT+ Goedel-Prover-V2-32B | ✗ | 97.9% | 862 | 449 | 4.0M | **1.2M** |

Table 3: **Ablation with/without retrieval.** HILBERT with retrieval achieves a higher pass rate while using less inference-time compute than without retrieval. Numbers show average calls and tokens per sample, computed over samples requiring subgoal decomposition.

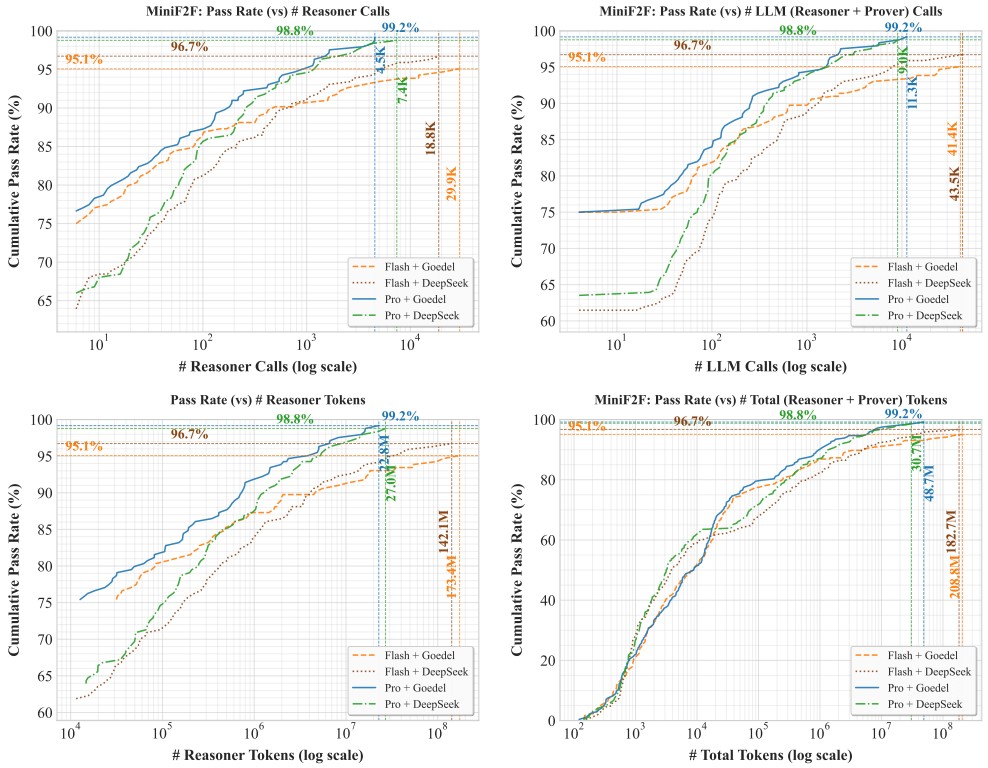

Figure 3: **Pass rate (vs) Inference-time Budget.** We plot the pass-rate for HILBERT on MiniF2F as a function of (top left) the number of Reasoner calls (top right) the total number of LLM (Reasoner + Prover) calls (bottom left) the number of tokens used by the Reasoner (bottom right) the total number of tokens used (Reasoner + Prover) per sample.

HILBERT achieves state-of-the-art performance on PutnamBench, solving 462 out of 660 problems (70.0% pass rate) when paired with Gemini 2.5 Pro. This surpasses the previous best method, the proprietary SeedProver (50.4%), by nearly 20 percentage points. Using the open-source `gpt-oss-120b` as the informal reasoner yields 88/660 problems solved (13.3%), performing comparably to Goedel-Prover-V2-32B with self-correction (86/644, 13.4%). HILBERT, with Gemini 2.5 Pro solves over 5 times more problems than the closest publicly available baseline, Goedel-Prover-V2-32B. We attribute this success to HILBERT's ability to compose long proofs (see Figure 9) without the long-context reasoning issues that plague traditional LLMs (Zhou et al., 2025a). For more information about the failure modes of HILBERT on PutnamBench, refer to Appendix A.7.

## 4.2 SCALING BEHAVIOR WITH INFERENCE-TIME COMPUTE

Unlike traditional prover LLMs that distribute compute across many independent proof attempts from scratch, HILBERT allocates inference-time compute across multiple interconnected stages, from subgoal decomposition to subgoal proof generation. Since this compute allocation is adaptive, it cannot be captured by a simple count of independent attempts. To illustrate the compute-performance tradeoff, we plot HILBERT's pass rate against the per-sample number of calls to (1) the Reasoner and (2) the Reasoner + Prover combined (Figure 3). The results reveal a clear scaling relationship where pass rates increase with the number of calls per sample. Our best-performing configuration (Gemini 2.5 Pro with Goedel Prover) requires at most 4.5K reasoner calls and 11.3K total calls, significantly fewer than DeltaProver's 16,384 calls with Gemini 2.5 Pro. Interestingly, the weaker reasoner (Gemini 2.5 Flash) demands a substantially higher inference budget to achieve comparable performance with both prover variants. While HILBERT+ DeepSeek Prover starts with lower pass rates, it demonstrates faster improvement rates, particularly in low-budget settings, eventually matching HILBERT+Goedel-Prover performance.

We also plot the pass-rate with token usage and observe a continuous increase in pass rate as token usage increases. Notably, the most challenging problems required 22.8M and 27.0M tokens for

the Gemini 2.5 Pro variants with Goedel-Prover-V2 and DeepSeek-Prover-V2, respectively. These token counts far exceed the context length of most LLMs, demonstrating that our agentic framework enables models to go beyond their inherent context limitations when solving complex mathematical problems, at the cost of increased inference-time computation. For additional analyses of pass rates versus prover/verifier calls and inference-time compute behavior on PutnamBench, refer to Appendix A.6.

## 4.3 ABLATION STUDIES

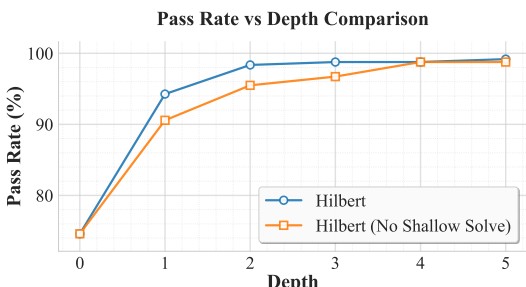

Figure 4: Pass rate (vs) recursive depth $D$ on MiniF2F for HILBERT (Gemini 2.5 Pro) + Goedel-Prover-V2-32B

**Performance (vs) depth.** To evaluate the effectiveness of subgoal decomposition, we analyze the pass rate of HILBERT using Gemini 2.5 Pro + Goedel-Prover-V2-32B on the MiniF2F dataset across different recursive depths $D$. The baseline ($D = 0$) corresponds to no decomposition, where we report the standalone Prover (pass@4) performance. We compare two configurations: the full HILBERT system, and a variant with shallow solving disabled ($K_{\text{informal passes}} = 0$). This variant relies solely on using the Prover for resolving subgoals. Figure 4 shows performance across different values of $D$, and demonstrates substantial gains from subgoal decomposition. Both configurations show monotonically increasing performance with depth, but exhibit different convergence patterns. The full HILBERT system achieves rapid performance gains, reaching 98.36% at $D = 2$ and 98.7% by $D = 3$. In contrast, the no-shallow-solve variant requires greater depth to achieve comparable performance. The consistent improvement over the $D = 0$ baseline (75% pass rate) validates the efficacy of hierarchical subgoal decomposition, with the full system achieving near-optimal performance at relatively shallow depths.

**Retrieval Ablation.** To assess the impact of the Retriever on both performance and computational efficiency, we compare HILBERT to a variant that omits the retrieval step. We experiment on MiniF2F across two Prover configurations: DeepSeek-Prover-V2-7B and Goedel-Prover-V2-32B. Table 3 presents the results. With retrieval enabled, HILBERT achieves higher pass rates across both configurations: 98.4% vs 97.1% for DeepSeek Prover and 99.2% vs 97.9% for Goedel Prover. More importantly, retrieval significantly reduces inference-time compute utilitzation. For the DeepSeek model, retrieval decreases reasoner calls from 426 to 420, average prover calls from 290 to 205, and average reasoner tokens from 2.1M to 1.9M. The efficiency gains are even more pronounced for HILBERT with Goedel Prover V2, where retrieval reduces average reasoner calls from 862 to 548 and average reasoner tokens from 4.0M to 2.3M. These results show that retrieval improves both performance and efficiency by surfacing useful theorems that simplify proofs and prevent failures from incorrect theorem names.

## 5 CONCLUSION

We present HILBERT, a hierarchical agentic framework that bridges formal theorem proving in Lean with the informal mathematical reasoning capabilities of general-purpose LLMs. Our approach recursively decomposes complex problems into manageable subgoals and orchestrates informal reasoners (Gemini 2.5 Pro/Flash/`gpt-oss-120b`) with formal provers (DeepSeek-Prover-V2-7B and Goedel-Prover-V2-32B) to solve theorems that neither component can handle alone. HILBERT achieves state-of-the-art performance on miniF2F with pass rates of 90.8% to 99.2%. On the challenging PutnamBench dataset, HILBERT with Gemini 2.5 Pro achieves 70.0% pass rate, nearly 20 percentage points above previous methods and approaching the 82% informal proof rate reported in Dekoninck et al. (2025).

In the future, we plan to leverage this framework to train increasingly capable models. Proofs and reasoning traces generated by HILBERT can be used to train better Prover and Reasoner models.

These improved models should be able to solve more complex problems than before, resulting in a virtuous cycle that has the potential to continually advance mathematical reasoning capabilities.

## ACKNOWLEDGMENTS

SV would like to thank Bohan Lyu and Sharut Gupta for several useful discussions about this work. RY was supported in part by the U.S. Army Research Office under Army-ECASE award W911NF-07-R-0003-03, the U.S. Department Of Energy, Office of Science, IARPA HAYSTAC Program, NSF Grants #2205093, #2146343, and #2134274, CDC-RFA-FT-23-0069, as well as DARPA AIE FoundSci and DARPA YFA.

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

# A APPENDIX

## A.1 ALGORITHM

The complete algorithm is presented across multiple blocks for clarity and modularity. Algorithm 1 provides the main entry point and high-level control flow, while Algorithm 2 details the subgoal resolution strategies. Algorithms 3 and 4 focus on sketch generation, validation, and assembly processes. Algorithm 5 contains the core proof generation functions that interface with different LLM components, while Algorithm 6 specifies the prompt-based functions for various reasoning tasks. Algorithm 7 handles error correction and refinement procedures, and Algorithm 8 provides supporting functions for theorem retrieval and verification.

---

**Algorithm 1** HILBERT: Hierarchical Proof Generation System

---

 1: **function** GENERATEPROOF(problem, header)
 2:     ▷ *Input: problem (formal statement), header (context)*
 3:
 4:     ▷ **Phase 1:** Direct Proof Attempt
 5:     proof ← ATTEMPTPROVERLLMPROOF(problem, header)
 6:     **if** proof $\neq \perp$ **then**
 7:         **return** proof
 8:     **end if**
 9:
10:     ▷ **Phase 2:** Subgoal Decomposition
11:     proof ← SUBGOALDECOMPOSITION(problem, header, depth=1)
12:     **return** proof
13: **end function**
14:
15: **function** SUBGOALDECOMPOSITION(problem, header, depth)
16:     ▷ *Decompose problem into subgoals and solve recursively*
17:     **if** depth $> D$ **then**
18:         **return** $\perp$    ▷ *Maximum recursion depth reached*
19:     **end if**
20:
21:     **for** attempt ← 1 **to** $K_{\text{sketch attempts}}$ **do**
22:         relevant_theorems ← RETRIEVETHEOREMS(problem)
23:         sketch ← GENERATEPROOFSKETCH(problem, relevant_theorems)
24:         sketch_assembled, subgoals, proved_subgoals ←
         REFINEANDVALIDATESKETCH(problem, sketch, header, relevant_theorems)
25:
26:         **if** sketch_assembled $\neq \perp$ **then**
27:             final_proof ← SOLVEALLSUBGOALS(subgoals, proved_subgoals,
     sketch_assembled, header, depth)
28:             **if** final_proof $\neq \perp$ **then**
29:                 **return** final_proof
30:             **end if**
31:         **end if**
32:     **end for**
33:     **return** $\perp$
34: **end function**

---

---

**Algorithm 2** HILBERT: Subgoal Resolution

---

1: **function** SOLVEALLSUBGOALS(subgoals, proved_subgoals, sketch_assembled, header, depth)
2:     ▷ *Solve all remaining subgoals and assemble final proof*
3:     subgoal_proofs ← ∅
4:
5:     **for all** subgoal ∈ subgoals \ proved_subgoals **do**
6:         proof ← SOLVESUBGOAL(subgoal, header, depth)
7:         **if** proof = ⊥ **then**
8:             **return** ⊥     ▷ *Failed to prove required subgoal*
9:         **end if**
10:         subgoal_proofs[subgoal] ← proof
11:     **end for**
12:
13:     final_proof ← CONCATENATE(header, subgoal_proofs, sketch_assembled)
14:     **return** final_proof
15: **end function**
16:
17: **function** SOLVESUBGOAL(subgoal, header, depth)
18:     ▷ *Solve individual subgoal with multiple strategies*
19:
20:     **Strategy 1:** Direct Prover Attempt
21:     proof ← ATTEMPTPROVERLLMPROOF(subgoal, header)
22:     **if** proof ≠ ⊥ **then**
23:         **return** proof
24:     **end if**
25:
26:     **Strategy 2:** Shallow Solve with Reasoner
27:     relevant_theorems ← RETRIEVETHEOREMS(subgoal)
28:     proof ← SHALLOWSOLVE(subgoal, header, relevant_theorems)
29:     **if** proof ≠ ⊥ **then**
30:         **return** proof
31:     **end if**
32:
33:     **Strategy 3:** Recursive Decomposition
34:     **if** depth < $D$ **then**
35:         proof ← SUBGOALDECOMPOSITION(subgoal, header, depth + 1)
36:         **if** proof ≠ ⊥ **then**
37:             **return** proof
38:         **end if**
39:     **end if**
40:     **return** ⊥
41: **end function**

---

---

**Algorithm 3** HILBERT: Sketch Validation and Refinement

---

1: **function** REFINEANDVALIDATESKETCH(problem, sketch, header, relevant_theorems)
2:      ▷ *Iteratively refine sketch until all subgoals are valid*
3:    **for** correction ← 1 **to** $K_{\text{sketch corrections}}$ **do**
4:      sketch_syntactic ← COMPILEANDCORRECTSYNTAXERRORS(problem, sketch, header, relevant_theorems)
5:      **if** sketch_syntactic == ⊥ **then**
6:        **return** ⊥, ∅, ∅
7:      **end if**
8:      subgoals ← EXTRACTSUBGOALS(sketch_syntactic, header)
9:      **if** subgoals == ⊥ **then**
10:       **return** ⊥, ∅, ∅
11:      **end if**
12:      sketch_assembled ← ASSEMBLEPROOFFROMSUBGOALS(sketch_syntactic, subgoals, header)
13:      **if** sketch_assembled == ⊥ **then**
14:       **return** ⊥, ∅, ∅
15:      **end if**
16:      valid, verified_subgoals, proved_subgoals, error_justification ← VALIDATESUBGOALS(subgoals, header)
17:      **if** valid **then**
18:       **return** sketch_assembled, verified_subgoals, proved_subgoals
19:      **else**
20:       sketch ← REFINESKETCHBASEDONERROR(sketch_syntactic, error_justification)
21:      **end if**
22:    **end for**
23:    **return** ⊥, ∅, ∅
24: **end function**
25:
26: **function** VALIDATESUBGOALS(subgoals, header)
27:      ▷ *Validate subgoals through formal proving and correctness checking*
28:    verified_subgoals ← ∅
29:    proved_subgoals ← {}
30:
31:    **for all** subgoal ∈ subgoals **do**
32:      proof ← ATTEMPTPROVERLLMPROOF(subgoal, header)
33:      **if** proof ≠ ⊥ **then**
34:       verified_subgoals ← verified_subgoals ∪ {subgoal}
35:       proved_subgoals[subgoal] ← proof
36:      **else**
37:       mathematically_correct, justification ← CHECKMATHEMATICALCORRECTNESS(subgoal)
38:       **if** mathematically_correct **then**
39:        verified_subgoals ← verified_subgoals ∪ {subgoal}
40:       **else**
41:        **return** false, ∅, ∅, justification
42:       **end if**
43:      **end if**
44:    **end for**
45:    **return** true, verified_subgoals, proved_subgoals, ⊥
46: **end function**

---

---

**Algorithm 4** HILBERT: Proof Sketch Refinement and Assembly

---

1: **function** COMPILEANDCORRECTSYNTAXERRORS(problem, sketch, header, relevant_theorems)
2:   ▷ *Compile sketch with sorry statements and correct errors*
3:   verified, error_message ← VERIFYPROOF(header + sketch)
4:   **if** verified **then**
5:     **return** sketch
6:   **end if**
7:
8:   ▷ *Error correction loop for sketch*
9:   **for** correction ← 1 **to** $K_{\text{theorem corrections}}$ **do**
10:     augmented_theorems ← AUGMENTTHEOREMS(error_message, problem, relevant_theorems)
11:     sketch ← CORRECTSKETCHERROR(sketch, error_message, augmented_theorems)
12:     verified, error_message ← VERIFYPROOF(header + sketch)
13:     **if** verified **then**
14:       **return** sketch
15:     **end if**
16:   **end for**
17:   **return** ⊥
18: **end function**
19:
20: **function** ASSEMBLEPROOFFROMSUBGOALS(sketch, subgoals, header)
21:   ▷ *Assemble complete proof outline with verification*
22:   all_theorems ← CONCATENATETHEOREMS(subgoals)
23:   sketch_assembled ← REASONERLLM(USE_SKETCH_AND_THEOREMS_PROMPT, sketch, all_theorems)
24:   corrected_proof ← VERIFYANDCORRECTPROOFWITHTHEOREMS(sketch_assembled, all_theorems, header)
25:   **return** corrected_proof
26: **end function**
27:
28: **function** VERIFYANDCORRECTPROOFWITHTHEOREMS(sketch_assembled, theorems, header)
29:   ▷ *Verify assembled sketch and correct errors*
30:   full_proof ← header + theorems + sketch_assembled
31:   verified, error ← VERIFYPROOF(full_proof)
32:   **if** verified **then**
33:     **return** sketch_assembled
34:   **end if**
35:
36:   **for** correction ← 1 **to** $K_{\text{theorem corrections}}$ **do**
37:     sketch_assembled ← REASONERLLM(ASSEMBLY_CORRECTION_PROMPT, error)
38:     full_proof ← header + theorems + sketch_assembled
39:     verified, error ← VERIFYPROOF(full_proof)
40:     **if** verified **then**
41:       **return** sketch_assembled
42:     **end if**
43:   **end for**
44:   **return** ⊥
45: **end function**

---

---

**Algorithm 5** HILBERT: Proof Generation

---

1: **function** ATTEMPTPROVERLLMPROOF(problem, header)
2:        ▷ *Multiple attempts with formal prover LLM*
3:    **for** attempt ← 1 **to** $K_{\text{formal attempts}}$ **do**
4:      proof ← PROVERLLM(problem)
5:      verified, error ← VERIFYPROOF(header + proof)
6:      **if** verified **then**
7:        **return** proof
8:      **end if**
9:    **end for**
10:    **return** ⊥
11: **end function**
12:
13: **function** GENERATEPROOFSKETCH(problem, relevant_theorems)
14:        ▷ *Generate informal proof sketch using prompts*
15:    informal_proof ← REASONERLLM(INFORMAL_PROOF_PROMPT, problem, relevant_theorems)
16:    sketch ← REASONERLLM(CREATE_LEAN_SKETCH_PROMPT, problem, relevant_theorems, informal_proof)
17:    **return** sketch
18: **end function**
19:
20: **function** SHALLOWSOLVE(subgoal, header, relevant_theorems)
21:        ▷ *Shallow solve with error correction loop*
22:    proof ← ATTEMPTREASONERPROOF(subgoal, relevant_theorems)
23:    verified, error_message ← VERIFYPROOF(header + proof)
24:    **if** verified **then**
25:      **return** proof
26:    **end if**
27:
28:        ▷ *Error correction loop*
29:    **for** correction ← 1 **to** $K_{\text{subgoal corrections}}$ **do**
30:      augmented_theorems ← AUGMENTTHEOREMS(error_message, subgoal, relevant_theorems)
31:      proof ← CORRECTPROOFERROR(proof, error_message, augmented_theorems)
32:      verified, error_message ← VERIFYPROOF(header + proof)
33:      **if** verified **then**
34:        **return** proof
35:      **else**
36:        ▷ *Check proof length cutoff when verification fails*
37:        **if** $|\text{proof}| > K_{\text{max shallow solve length}}$ **then**
38:          **return** ⊥    ▷ *Proof too long and still incorrect, abandon*
39:        **end if**
40:      **end if**
41:    **end for**
42:    **return** ⊥
43: **end function**

---

---

**Algorithm 6** HILBERT: LLM Prompt Functions

---

1: **function** ATTEMPTREASONERPROOF(subgoal, relevant_theorems)
2:    ▷ *Shallow solve using informal reasoning*
3:   proof ← REASONERLLM(SOLVE_SUBGOAL_PROMPT, subgoal, relevant_theorems)
4:   **return** proof
5: **end function**
6:
7: **function** CHECKMATHEMATICALCORRECTNESS(subgoal)
8:    ▷ *Verify mathematical correctness of subgoal*
9:   correct, justification ← REASONERLLM(DETERMINE_IF_CORRECT_SUBGOAL_PROMPT, subgoal)
10:   **return** correct, justification
11: **end function**
12:
13: **function** EXTRACTSUBGOALS(sketch, header)
14:    ▷ *Extract have statements as independent subgoals*
15:   subgoals ← REASONERLLM(EXTRACT_SUBGOALS_FROM_SKETCH_PROMPT, sketch)
16:
17:    ▷ *Syntax check and correction for each subgoal*
18:   corrected_subgoals ← ∅
19:   **for all** subgoal ∈ subgoals **do**
20:     verified, error ← VERIFYPROOF(header + subgoal)
21:     **if** verified **then**
22:       corrected_subgoals ← corrected_subgoals ∪ {subgoal}
23:     **else**
24:        ▷ *Error correction loop*
25:       corrected ← false
26:       **for** attempt ← 1 **to** $K_{\text{subgoal error corrections}}$ **do**
27:         subgoal ← CORRECTTHEOREMERROR(subgoal, error)
28:         verified, error ← VERIFYPROOF(header + subgoal)
29:         **if** verified **then**
30:           corrected_subgoals ← corrected_subgoals ∪ {subgoal}
31:           corrected ← true
32:           **break**   ▷ *Successfully corrected*
33:         **end if**
34:       **end for**
35:       **if** ¬corrected **then**
36:         **return** ⊥   ▷ *Failed to correct subgoal, return failure*
37:       **end if**
38:     **end if**
39:   **end for**
40:
41:   **return** corrected_subgoals
42: **end function**

---

---

**Algorithm 7** HILBERT: Error Correction

---

1: **function** REFINESKETCHBASEDONERROR(sketch, error_justification)
2:          ▷ *Refine proof sketch based on subgoal validation errors*
3:      refined ← REASONERLLM(CORRECT_SKETCH_BASED_ON_INCORRECT_SUBGOAL_PROMPT, sketch, error_justification)
4:      **return** refined
5: **end function**
6:
7: **function** CORRECTSKETCHERROR(sketch, error_message, relevant_theorems)
8:          ▷ *Correct syntax and compilation errors*
9:      corrected ← REASONERLLM(PROOF_SKETCH_CORRECTION_PROMPT, error_message, sketch, relevant_theorems)
10:      **return** corrected
11: **end function**
12:
13: **function** CORRECTPROOFERROR(proof, error_message, augmented_theorems)
14:          ▷ *Correct proof errors using error feedback*
15:      corrected ← REASONERLLM(PROOF_CORRECTION_PROMPT, error_message, proof, augmented_theorems)
16:      **return** corrected
17: **end function**
18:
19: **function** CORRECTTHEOREMERROR(subgoal, error_message)
20:          ▷ *Correct syntax errors in extracted subgoals*
21:      corrected          ←          REASONERLLM(SUBGOAL_SYNTAX_CORRECTION_PROMPT, error_message, subgoal)
22:      **return** corrected
23: **end function**

---

---

**Algorithm 8** HILBERT: Retrieval and Helper Functions

---

1: **function** RETRIEVETHEOREMS(`problem`)
2:         ▷ *Theorem retrieval from Mathlib*
3:     **if** `retrieval_enabled` **then**
4:         `search_queries` ← GENERATESEARCHQUERIES(`problem`)
5:         `candidate_theorems` ← SEMANTICSEARCHENGINE(`search_queries`)
6:         `relevant_theorems` ← SELECTRELEVANTTHEOREMS(`candidate_theorems`, `problem`)
7:         **return** `relevant_theorems`
8:     **else**
9:         **return** $\emptyset$
10:     **end if**
11: **end function**
12:
13: **function** GENERATESEARCHQUERIES(`problem`)
14:         ▷ *Generate search queries for theorem retrieval*
15:     `queries` ← REASONERLLM(SEARCH_QUERY_PROMPT, `problem`)
16:     **return** `queries`
17: **end function**
18:
19: **function** SELECTRELEVANTTHEOREMS(`candidate_theorems`, `problem`)
20:         ▷ *Select most relevant theorems from candidates*
21:     `selected` ← REASONERLLM(SEARCH_ANSWER_PROMPT, `problem`, `candidate_theorems`)
22:     **return** `selected`
23: **end function**
24:
25: **function** VERIFYPROOF(`full_proof`)
26:         ▷ *Verify proof using Lean verifier*
27:     `result, error_message` ← LEANVERIFIER(`full_proof`)
28:     **return** `result, error_message`
29: **end function**
30:
31: **function** AUGMENTTHEOREMS(`error_message`, `problem`, `existing_theorems`)
32:         ▷ *Add theorems for missing identifiers*
33:     `missing_ids` ← EXTRACTMISSINGIDENTIFIERS(`error_message`)
34:     **if** `missing_ids` $\neq \emptyset$ **then**
35:         `additional_theorems` ← RETRIEVETHEOREMS(`problem` + `missing_ids`)
36:         **return** `existing_theorems` + `additional_theorems`
37:     **end if**
38:     **return** `existing_theorems`
39: **end function**

---

## A.2 PROMPTS

---

**Search Query Generation (`SEARCH_QUERY_PROMPT`)**

```
You are helping solve a Lean theorem proving problem using the mathlib library.
Before attempting to write the proof, you must first search for relevant theorems and
↪  tactics.

Search Process:
1. Identify key concepts: Break down the problem into mathematical concepts, operations,
↪  and structures involved.
2. Generate search queries: For each concept, create informal search strings that
↪  describe:
   - Relevant theorems or results (e.g., "associativity of addition", "existence of
   ↪  inverse elements")
   - Useful tactics (e.g., "simplify arithmetic expressions", "split conjunctions")
   - Properties (e.g., "group structure on integers", "metric space properties")
   - Relevant definitions useful for the proof or any used theorem (e.g. "definition of
   ↪  a group", "definition of a metric space")

Search Query Format:
Enclose each search query in <search> tags with your informal description. Limit
↪  yourself to a maximum of 5 search queries. Make the search queries simple, concise,
↪  and clear.

Guidelines:
- You can either search by theorem name or natural language description
- Search for theorems that might automate parts of the proof
- Consider edge cases and special conditions mentioned in the problem

Problem to Solve:
{problem}
```

---

**Theorem Selection (`SEARCH_ANSWER_PROMPT`)**

```
You are helping to solve a Lean theorem proving problem using the mathlib library. The problem is:
{problem}

Here are some potentially relevant theorems and definitions:
{theorems}

Instructions:
1. Select important theorems and definitions necessary to solve the problem.
2. IMPORTANT: ONLY SELECT theorems from the GIVEN list.
3. Enclose each of them in separate <theorem> tags.
4. Only state the full names of the theorems. Do NOT include the module name.
5. Select all theorems that could be useful in the intermediate steps of the proof.
```

---

**Informal Proof Generation (`INFORMAL_PROOF_PROMPT`)**

```
You are a mathematical expert whose goal is to solve problems with rigorous
mathematical reasoning.

{useful_theorems_section}
Instructions:
1. Provide a natural language, step-by-step proof for the given problem.
2. Start from the given premises and reason step-by-step to reach the conclusion.
3. Number each step of the proof as 1, 2, and so on.
4. Be as pedantic and thorough as possible.
5. Keep each step precise, increase the number of steps if needed.
6. Do NOT gloss over any step. Make sure to be as thorough as possible.
7. Show the explicit calculations/simplifications, theorem applications and case
   analysis.
8. Enclose the informal proof in <informal_proof> tags.

Problem Statement: {problem}
```

## Lean Sketch Creation (`CREATE_LEAN_SKETCH_PROMPT`)

```
You are a Lean 4 expert who is trying to help write a proof in Lean 4.

Problem Statement: {problem}

{useful_theorems_section}
Informal Proof:
{informal_proof}

Instructions:

Use the informal proof to write a proof sketch for the problem in Lean 4 following
these guidelines:
- Break complex reasoning into logical sub-goals using `have` statements.
- The subgoals should build up to prove the main theorem.
- Make sure to include all the steps and calculations from the given proof in the
  proof sketch.
- Each subgoal should ideally require applying just one key theorem or lemma, or a
  few tactic applications.
- Base subgoals around:
  - Useful theorems mentioned in the problem context
  - Standard library theorems (like arithmetic properties, set operations, etc.)
  - The supplied premises in the theorem statement
- Do NOT create subgoals identical to any of the given hypotheses
- Do NOT create subgoals that are more complex than the original problems. The
  subgoals should be SIMPLER than the given problem.
- Do NOT skip over any steps. Do NOT make any mathematical leaps.

**Subgoal Structure Requirements:**
- **Simplicity**: Each subgoal proof should be achievable with 1-3 basic tactics
- **Atomic reasoning**: Avoid combining multiple logical steps in one subgoal
- **Clear progression**: Show logical flow: `premises → intermediate steps → final result`
- **Theorem-focused**: Design each subgoal to directly apply a specific theorem when possible

NOTE: Only add sub-goals that simplify the proof of the main goal.

When writing Lean proofs, maintain consistent indentation levels.

Rules:
1. Same proof level = same indentation: All tactics at the same logical level must
   use identical indentation
2. Consistent characters: Use either tabs OR spaces consistently (don't mix)
3. Proper nesting: Indent sub-proofs one level deeper than their parent
4. Do NOT nest `have` statements in each other. Use distinct sub-goals as much as
   possible. Ensure all sub goals are named. Do NOT create anonymous have statements.
5. Do NOT include any imports or open statements in your code.
6. One line = One `have` subgoal. Do NOT split subgoals across different lines.
7. Use proper Lean 4 syntax and conventions. Ensure the proof sketch is enclosed in
   triple backticks ```lean```
8. Use `sorry` for all subgoal proofs - focus on structure, not implementation
9. **Do NOT use `sorry` for the main goal proof** - use your subgoals to prove it
10. NEVER use `sorry` IN the theorem statement itself
11. Ensure subgoals collectively provide everything needed for the main proof
12. Make the logical dependencies between subgoals explicit. Ensure that the subgoals
    are valid and provable in Lean 4.
13. Do NOT change anything in the original theorem statement.

Lean Hints:
{lean_hints}
```

## Lean Sketch Creation (`CREATE_LEAN_SKETCH_PROMPT`) (continued)

```
IMPORTANT INSTRUCTION: Do NOT, under ANY circumstances, allow division and
subtraction operations on natural number literals with UNDEFINED types, unless
REQUIRED by the theorem statement. For example, do NOT allow literals like `1 / 3`
or `2 / 5` or `1 - 3` ANYWHERE in ANY of the subgoals. ALWAYS specify the types.
AVOID natural number arithmetic UNLESS NEEDED by the theorem statement.
ALWAYS specify types when describing fractions. For example, ((2 : ℝ) / 3) or
((2 : ℚ) / 3) instead of (2 / 3). Do this everywhere EXCEPT the given theorem statement.
IMPORTANT INSTRUCTION: Do NOT, under ANY circumstances, allow division and
subtraction operations on variables of type natural numbers (Nat or ℕ), unless
REQUIRED by the theorem statement. For example, do NOT allow expressions like (a-b)
or (a/b) where a, b are of type ℕ. ALWAYS cast the variables to a suitable type
(ℤ, ℚ or ℝ) when performing arithmetic operations. AVOID natural number arithmetic
UNLESS NEEDED by the theorem statement.
```

## Subgoal Extraction (`EXTRACT_SUBGOALS_FROM_SKETCH_PROMPT`)

```
From this proof sketch, extract any missing proofs (specified with `sorry`) as
independent subgoals (theorems).
Instructions:
1. Use the same name as the have statements for the theorems.
2. Each subgoal should have the relevant context from the previous subgoals needed
   to simplify the proof as much as possible.
3. There should be as many extracted theorems as `sorry`s in the given theorem.
4. Do NOT include any imports or open statements. Do NOT add any definitions. ONLY
   include the theorem statement.
5. Use a separate Lean 4 ``lean`` block for each subgoal.
6. Use sorry for the proof. Do NOT prove any theorem.
7. Do NOT change the conclusion of the theorems from the extracted subgoals. Keep
   them AS IT IS.
8. Do NOT change the conclusions of the preceding theorems when presenting them as
   hypotheses for the next subgoals. Keep them AS IT IS.
9. Do NOT duplicate theorem names. Use distinct theorem names for the different theorems.
10. Make sure the names and types of the premises/arguments in the extracted theorems
    MATCH the subgoals from which they are extracted.

IMPORTANT INSTRUCTION: Do NOT, under ANY circumstances, allow division and
subtraction operations on natural number literals with UNDEFINED types, unless
REQUIRED by the theorem statement. For example, do NOT allow literals like `1 / 3`
or `2 / 5` or `1 - 3` ANYWHERE in the theorem statement. ALWAYS specify the types.
AVOID natural number arithmetic UNLESS NEEDED by the theorem statement.
ALWAYS specify types when describing fractions. For example, ((2 : ℝ) / 3) or
((2 : ℚ) / 3) instead of (2 / 3)
IMPORTANT INSTRUCTION: Do NOT, under ANY circumstances, allow division and
subtraction operations on variables of type natural numbers (Nat or ℕ), unless
REQUIRED by the theorem statement. For example, do NOT allow expressions like (a-b)
or (a/b) where a, b are of type ℕ. ALWAYS cast the variables to a suitable type
(ℤ, ℚ or ℝ) when performing arithmetic operations. AVOID natural number arithmetic
UNLESS NEEDED by the theorem statement.

Lean Hints:
{lean_hints}

Proof Sketch:
```lean4
{proof_sketch}
```
```

## Subgoal Solving (`SOLVE_SUBGOAL_PROMPT`)

```
Think step-by-step to complete the following Lean 4 proof.

{problem}

Lean Hints:
{lean_hints}

Tactic Hints:
{tactic_hints}

Rules:
1. Same proof level = same indentation: All tactics at the same logical level must
   use identical indentation
2. Consistent characters: Use either tabs OR spaces consistently (don't mix)
3. Proper nesting: Indent sub-proofs one level deeper than their parent
4. Do NOT include any imports or open statements.
5. Use proper Lean 4 syntax and conventions. Ensure the proof sketch is enclosed in
   triple backticks ```lean```.
6. Only include a single Lean 4 code block, corresponding to the proof along with
   the theorem statement.
7. When dealing with large numerical quantities, avoid explicit computation as much
   as possible. Use tactics like rw to perform symbolic manipulation rather than
   numerical computation.
8. Do NOT use sorry.
9. Do NOT change anything in the original theorem statement.
{useful_theorems_section}
```

## Mathematical Correctness Check (`DETERMINE_IF_CORRECT_SUBGOAL_PROMPT`)

```
You are an expert in mathematics.

Your task is to evaluate whether the given mathematical theorem statement is
mathematically correct. You do NOT have to provide a proof for the theorem in Lean.

Evaluation criteria:
1. Mathematical validity: Check for logical errors, incorrect assumptions, or
   calculation mistakes.
2. Do NOT flag general results or helper lemmas that are true independent of the
   given premises. ONLY flag inaccuracies or mistakes.
5. Provability: Determine if the statement can be proven given the provided premises,
   or otherwise.

Assumptions:
1. The given premises are mathematically correct. Do NOT check this.
2. The syntax is guaranteed to be correct (do not assess syntax)

Theorem Statement:
{problem}

Report your answer as either:
• YES - if the statement is mathematically correct
• NO - if the statement has mathematical errors that prevent proof

Also provide a brief justification for your decision in <justification></justification>
tags, adding details about why the statement is correct or incorrect.
If it is incorrect, also provide a description of how the error can be corrected.
If there are missing arguments, make sure to add the relevant missing proof steps.
```

## Sketch Assembly (`USE_SKETCH_AND_THEOREMS_PROMPT`)

```
You are a Lean 4 expert. Your goal is to write a proof in Lean 4, according to the
given proof sketch, using the supplied theorems.

Proof sketch:
{proof_sketch}

Theorems:
{theorems_string}

Instructions:
1. You can assume that the theorems are correct and use them directly in your proof.
2. Do NOT modify the given theorems.
3. Do NOT prove the given theorems.
4. Do NOT modify the given proof sketch steps. Simply apply the given theorems to
   complete the missing `sorry` steps.
5. Do NOT use `sorry` in your proof.
6. Do NOT include any imports or definitions or open statements.
7. Do NOT re-define the given theorems in your response.
8. Do NOT write a proof for any subgoal from scratch. ALWAYS use the supplied theorems.

IMPORTANT INSTRUCTION: Do NOT, under ANY circumstances, allow division and
subtraction operations on natural number literals with UNDEFINED types, unless
REQUIRED by the theorem statement. For example, do NOT allow literals like `1 / 3`
or `2 / 5` or `1 - 3`. ALWAYS specify the types. AVOID natural number arithmetic
UNLESS NEEDED by the theorem statement.
ALWAYS specify types when describing fractions. For example, ((2 : ℝ) / 3) or
((2 : ℚ) / 3) instead of (2 / 3). Do this everywhere EXCEPT the given theorem statement.
IMPORTANT INSTRUCTION: Do NOT, under ANY circumstances, allow division and
subtraction operations on variables of type natural numbers (Nat or ℕ), unless
REQUIRED by the theorem statement. For example, do NOT allow expressions like (a-b)
or (a/b) where a, b are of type ℕ. ALWAYS cast the variables to a suitable type
(ℤ, ℚ or ℝ) when performing arithmetic operations. AVOID natural number arithmetic
UNLESS NEEDED by the theorem statement.

Your answer should be a single Lean 4 block containing the completed proof for the
given theorem.
```

## Assembly Correction (`ASSEMBLY_CORRECTION_PROMPT`)

```
The following Lean 4 code has compilation errors. Please fix the errors while
maintaining the mathematical meaning.

{error_message}

Lean Hints:
{lean_hints}

Instructions:
1. Analyze what the theorem is trying to prove. Then, analyze why the error is
   happening, step-by-step. Add a brief explanation.
2. Then, provide a corrected version of the Lean 4 code that addresses these
   specific errors.
3. You should ONLY correct the main theorem that appears at the end. Do NOT
   change any of the helper theorems.
3. Do NOT include any other Lean code blocks except for the proof. Do NOT
   include any imports or open statements.
4. Do NOT use `sorry` in any part of the proof.
5. Do NOT change anything in the original theorem statement.
6. Do NOT include the helper theorem definitions in your response.
7. Do NOT write a proof for any subgoal from scratch. ALWAYS use the supplied
   theorems.
```

## Sketch Refinement Based on Incorrect Subgoal (`CORRECT_SKETCH_BASED_ON_INCORRECT_SUBGOAL_PROMPT`)

```
You are an expert in writing Lean 4 proofs. You are given a Lean 4 proof sketch
where one of the subgoals has some issues.
Your task is to fix the issues and write a new proof sketch.

Proof Sketch:
{proof_sketch}

Issues:
{issues}

Lean Hints:
{lean_hints}

Rules:
1. Same proof level = same indentation: All tactics at the same logical level
   must use identical indentation
2. Consistent characters: Use either tabs OR spaces consistently (don't mix)
3. Proper nesting: Indent sub-proofs one level deeper than their parent
4. Do NOT nest `have` statements in each other. Write different have statements
   for different sub goals.
5. Ensure all sub goals are named. Do NOT create anonymous have statements.
6. Do NOT include any imports or open statements.
7. One line = One `have` subgoal. Do NOT split subgoals across different lines.
8. Use proper Lean 4 syntax and conventions. Ensure the proof sketch is enclosed
   in triple backticks ```lean```
9. Use `sorry` for all subgoal proofs - focus on structure, not implementation
10. **Do NOT use `sorry` for the main goal proof** - use your subgoals to prove it
11. NEVER use `sorry` IN the theorem statement itself
12. Ensure subgoals collectively provide everything needed for the main proof
13. Make the logical dependencies between subgoals explicit. Ensure that the
    subgoals are valid and provable in Lean 4.
14. Modify only the incorrect subgoal and everything that follows it in the proof
    sketch. Leave all preceding portions unchanged.
15. Either modify the problematic subgoals to fix the errors, or add additional
    subgoals to fill in the missing mathematical arguments.
```

**Proof Sketch Correction (`PROOF_SKETCH_CORRECTION_PROMPT`)**

```
The following Lean 4 code has compilation errors. Please fix the errors while
maintaining the mathematical meaning.

Original statement: {informal_statement}

{error_message}

Lean Hints:
{lean_hints}

Instructions:
1. Analyze what the theorem is trying to prove. Then, analyze why the error is
   happening, step-by-step. Add a brief explanation.
2. Then, provide a corrected version of the Lean 4 code that addresses these
   specific errors.
3. Do NOT include any other Lean code blocks except for the proof. Do NOT
   include any imports or open statements.
4. Use sorry for the proof of all `have` statements.
5. Ensure there are no use of `sorry` statements outside of `have` statements.
   Do NOT use `sorry` while proving the main theorem.
6. Do NOT change anything in the original theorem statement.
7. Do NOT nest `have` statements in each other. Use distinct sub-goals as much
   as possible. Ensure all sub goals are named. Do NOT create anonymous have
   statements.

{useful_theorems_section}
```

**Proof Correction (`PROOF_CORRECTION_PROMPT`)**

```
The following Lean 4 code has compilation errors. Please fix the errors while
maintaining the mathematical meaning.

{error_message}

Instructions:
1. Analyze what the theorem is trying to prove. Then, analyze why the error is
   happening, step-by-step. Add a brief explanation.
2. Then, provide a corrected version of the Lean 4 code that addresses these
   specific errors.
3. Do NOT include any other Lean code blocks except for the proof.
4. Do NOT use sorry.
5. Do NOT include any imports or open statements.
6. Do NOT change anything in the original theorem statement.

{useful_theorems_section}
```

**Subgoal Syntax Correction (`SUBGOAL_SYNTAX_CORRECTION_PROMPT`)**

```
The following Lean 4 theorem has compilation errors. Please fix the errors while
maintaining the mathematical meaning.

{error_message}
Instructions:
1. Analyze why the error is happening, step-by-step. Add a brief explanation.
2. Then, provide a corrected version of the Lean 4 code that addresses these
   specific errors.
3. Do NOT include any other Lean code blocks except for the theorem.
4. Use sorry for the proof.
5. Do NOT include any imports or open statements.
{potentially_useful_theorems}
```

## A.3 IMPLEMENTATION DETAILS

We improve HILBERT's efficiency through several runtime optimizations focused on parallelization. The Prover LLM is served using vLLM (Kwon et al., 2023) and the Lean Verifier using Kimina Lean Server (Santos et al., 2025) to handle multiple requests in parallel.

We implement `AsyncJobPool`, a mechanism built around Python's `asyncio` library, to orchestrate parallel requests across our framework's multiple steps. Submitted jobs run concurrently until

specific completion criteria are met based on the algorithm step. Concurrency is controlled using Semaphores. We implement three completion criteria:

- **Wait for All**. The execution terminates when all jobs in the pool have finished execution. This criterion is used to parallelize across examples, and across subgoals (Section 3.2.2).
- **First-Success Termination**. Execution terminates as soon as one successful job is found, and pending jobs are terminated. This criterion is used to parallelize across proof attempts (the initial Prover attempts, and Steps 1 and 3 in Section 3.2.2).
- **First Failure**. Execution halts upon the first job failure, immediately canceling remaining jobs. This criterion is applied during subgoal correctness verification (Step 2 in Section 3.2.2). Since verification failures often indicate fundamental issues with the proof sketch that affect multiple subgoals, early termination prevents wasted computation on dependent subgoals, which may change after correcting the problematic subgoal.

## A.4 SUBGOAL DECOMPOSITION EXAMPLE

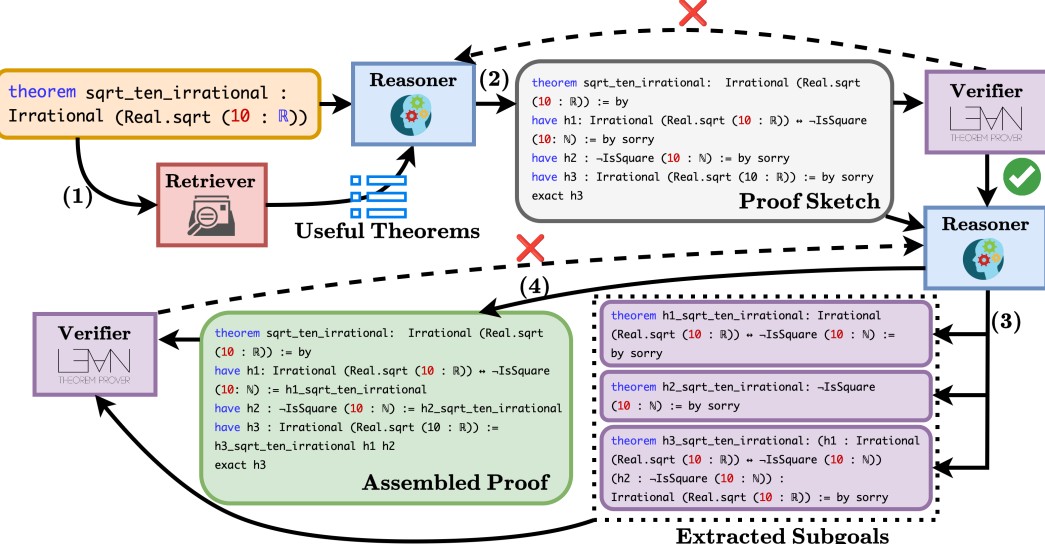

Figure 5: **Subgoal Decomposition Example.** We illustrate the subgoal decomposition process using the input theorem `sqrt_ten_irrational`. The process consists of four main steps: (1) We retrieve relevant theorems from Mathlib to inform the proof strategy. (2) The Reasoner generates a proof sketch, which is verified by the Lean Verifier for validity. If verification fails, error messages guide the Reasoner to make corrections. (3) The Reasoner extracts subgoals from the validated sketch and verifies their correctness, refining them as needed. (4) The Reasoner assembles a complete proof by incorporating the extracted subgoals into the original sketch. Since the subgoals lack proofs at this stage, they are denoted by `sorry`. This assembled proof undergoes final verification. The process outputs both the complete assembled proof and the verified subgoals (without their proofs). Note that while Steps (3) and (4) are shown together in this figure for simplicity, they represent distinct operations as detailed in Figure 2.

## A.5 MINIF2F PROOF EXAMPLE

We include an example to demonstrate how our hierarchical algorithm can help discover proofs that are too complex for either the Prover or informal Reasoner alone. We present the proof for Problem 4 from AMC 12B 2002, a problem which was unsolved even by DeepSeek Prover V2-671B after 8196 passes. We find that HILBERT is able to solve the problem with one layer of decomposition using Goedel Prover V2-32B model as the Prover.

```
import Mathlib
import Aesop

set_option maxHeartbeats 0
```

```
open BigOperators Real Nat Topology Rat
theorem k_one_amc12b_2002_p4 (n : ℕ) (h₀ : 0 < n) (k : ℕ) (hk : 41 * n + 42 = k * (42 *
↪ n)) (h_k_is_one : ∀ (k : ℕ), 41 * n + 42 = k * (42 * n) → k = 1) : k = 1 := by
  have h_main : k = 1 := by
    have h₁ : k = 1 := h_k_is_one k hk
    exact h₁

  exact h_main

theorem h_sum_is_frac_amc12b_2002_p4 (n : ℕ) (h₀ : 0 < n) : (1 / 2 + 1 / 3 + 1 / 7 + 1 /
↪ ↑n : ℚ) = (↑(41 * n + 42)) / ↑(42 * n) := by
  have h₁ : (1 / 2 + 1 / 3 + 1 / 7 + 1 / ↑n : ℚ) = (↑(41 * n + 42)) / ↑(42 * n) := by
    have h₂ : (n : ℚ) ≠ 0 := by
      norm_cast
      <;> linarith
    field_simp [h₂]
    <;> ring_nf
    <;> norm_cast
    <;> field_simp [h₂]
    <;> ring_nf
    <;> norm_cast
    <;> simp [mul_assoc]
    <;> norm_num
    <;> linarith

  apply h₁

theorem h_k_is_one_amc12b_2002_p4 (n : ℕ) (h₀ : 0 < n) : ∀ (k : ℕ), 41 * n + 42 = k * (42
↪ * n) → k = 1 := by
  intro k hk
  have h₁ : k ≠ 0 := by
    by_contra h
    have h₂ : k = 0 := by simpa using h
    rw [h₂] at hk
    have h₃ : 41 * n + 42 = 0 := by
      ring_nf at hk ⊢
      <;> nlinarith
    have h₄ : 41 * n + 42 > 0 := by
      nlinarith
    linarith

  have h₂ : k < 2 := by
    by_contra h
    have h₃ : k ≥ 2 := by
      omega
    have h₄ : k * (42 * n) ≥ 2 * (42 * n) := by
      have h₅ : k * (42 * n) ≥ 2 * (42 * n) := by
        have h₆ : k ≥ 2 := h₃
        have h₇ : (42 : ℕ) * n > 0 := by positivity
        nlinarith
      exact h₅
    have h₅ : 2 * (42 * n) = 84 * n := by
      ring
    have h₆ : k * (42 * n) ≥ 84 * n := by
      linarith
    have h₇ : 41 * n + 42 < 84 * n := by
      have h₈ : n ≥ 1 := by linarith
      have h₉ : 43 * n ≥ 43 := by
        nlinarith
      have h₁₀ : 43 * n - 42 ≥ 1 := by
        have h₁₁ : 43 * n ≥ 43 := by nlinarith
        omega
      have h₁₁ : 84 * n > 41 * n + 42 := by
        cases n with
        | zero => contradiction
        | succ n =>
          simp [Nat.mul_add, Nat.add_mul, Nat.add_assoc] at h₆ ⊢
          <;> ring_nf at h₆ ⊢
          <;> (try omega)
          <;> (try nlinarith)
        omega
      have h₈ : 41 * n + 42 < k * (42 * n) := by
        linarith
      linarith

  have h₃ : k = 1 := by
    have h₄ : k ≤ 1 := by
      omega
```

```
    have h₅ : k ≥ 1 := by
      by_contra h₆
      have h₇ : k = 0 := by
        omega
      contradiction
    omega

  exact h₃

theorem h_divides_amc12b_2002_p4 (n : ℕ) (h₀ : 0 < n) (h₁ : (1 / 2 + 1 / 3 + 1 / 7 + 1 /
↪  ↑n : ℚ).den = 1) (h_sum_is_frac : (1 / 2 + 1 / 3 + 1 / 7 + 1 / ↑n : ℚ) = (↑(41 * n +
↪  42)) / ↑(42 * n)) : 42 * n | 41 * n + 42 := by
  have h₂ : ((↑(41 * n + 42) : ℚ) / ↑(42 * n)).den = 1 := by
    have h₂₁ : ((↑(41 * n + 42) : ℚ) / ↑(42 * n)) = (1 / 2 + 1 / 3 + 1 / 7 + 1 / ↑n : ℚ)
    ↪  := by
      rw [h_sum_is_frac]
      <;> field_simp <;> ring_nf <;> norm_cast
      <;> field_simp <;> ring_nf <;> norm_cast
    rw [h₂₁]
    exact h₁

  have h₃ : (42 * n : ℤ) | (41 * n + 42 : ℤ) := by
    have h₃₁ : (((41 * n + 42 : ℤ) : ℚ) / (42 * n : ℤ)).den = 1 := by
      norm_cast at h₂ ⊢
      <;> simpa [div_eq_mul_inv] using h₂
    have h₃₂ : ∃ (k : ℤ), ((41 * n + 42 : ℤ) : ℚ) / (42 * n : ℤ) = (k : ℚ) := by
      have h₃₃ : (((41 * n + 42 : ℤ) : ℚ) / (42 * n : ℤ)).den = 1 := h₃₁
      have h₃₄ : ∃ (k : ℤ), ((41 * n + 42 : ℤ) : ℚ) / (42 * n : ℤ) = (k : ℚ) := by
        have h₃₅ : ∃ (k : ℤ), ((41 * n + 42 : ℤ) : ℚ) / (42 * n : ℤ) = (k : ℚ) := by
          have h₃₆ : (((41 * n + 42 : ℤ) : ℚ) / (42 * n : ℤ)).den = 1 := h₃₁
          have h₃₇ : ∃ (k : ℤ), ((41 * n + 42 : ℤ) : ℚ) / (42 * n : ℤ) = (k : ℚ) := by
            refine' ⟨(((41 * n + 42 : ℤ) : ℚ) / (42 * n : ℤ)).num, _⟩
            have h₃₈ : (((41 * n + 42 : ℤ) : ℚ) / (42 * n : ℤ)) = ((((41 * n + 42 : ℤ) :
            ↪  ℚ) / (42 * n : ℤ)).num : ℚ) := by
              have h₃₉ : (((41 * n + 42 : ℤ) : ℚ) / (42 * n : ℤ)).den = 1 := h₃₁
              have h₄₀ : (((41 * n + 42 : ℤ) : ℚ) / (42 * n : ℤ)) = ((((41 * n + 42 : ℤ) :
              ↪  ℚ) / (42 * n : ℤ)).num : ℚ) := by
                rw [← Rat.num_div_den (((41 * n + 42 : ℤ) : ℚ) / (42 * n : ℤ))]
                <;> field_simp [h₃₉]
                <;> norm_cast
                <;> simp_all [Rat.den_nz]
              exact h₄₀
            exact h₃₈
          exact h₃₇
        exact h₃₅
      exact h₃₄
    obtain ⟨k, h₃₃⟩ := h₃₂
    have h₃₄ : (42 * n : ℤ) | (41 * n + 42 : ℤ) := by
      have h₃₅ : ((41 * n + 42 : ℤ) : ℚ) / (42 * n : ℤ) = (k : ℚ) := h₃₃
      have h₃₆ : (42 * n : ℤ) ≠ 0 := by
        have h₃₇ : (n : ℕ) > 0 := h₀
        have h₃₈ : (42 * n : ℤ) > 0 := by
          norm_cast
          <;> nlinarith
        linarith
      have h₃₇ : ((41 * n + 42 : ℤ) : ℚ) = (k : ℚ) * (42 * n : ℤ) := by
        field_simp [h₃₆] at h₃₅ ⊢
        <;> ring_nf at h₃₅ ⊢ <;> norm_cast at h₃₅ ⊢
        <;> (try norm_num at h₃₅ ⊢)
        <;> (try linarith)
        <;> (try simp_all [mul_comm])
        <;> (try ring_nf at h₃₅ ⊢ <;> norm_cast at h₃₅ ⊢ <;> linarith)
        <;> (try { nlinarith })
        <;> (try {
          field_simp at h₃₅ ⊢
          <;> ring_nf at h₃₅ ⊢
          <;> norm_cast at h₃₅ ⊢
          <;> linarith
        })
      have h₃₈ : (41 * n + 42 : ℤ) = k * (42 * n : ℤ) := by
        norm_cast at h₃₇ ⊢
        <;> (try ring_nf at h₃₇ ⊢)
        <;> (try norm_num at h₃₇ ⊢)
        <;> (try linarith)
        <;> (try {
          field_simp at h₃₇ ⊢
          <;> ring_nf at h₃₇ ⊢
          <;> norm_cast at h₃₇ ⊢
          <;> linarith
```

```
      })
    have h₃₉ : (42 * n : ℤ) | (41 * n + 42 : ℤ) := by
      use k
      <;> linarith
    exact h₃₉
  exact h₃₄

have h₄ : 42 * n | 41 * n + 42 := by
  have h₄₁ : (42 * n : ℤ) | (41 * n + 42 : ℤ) := h₃
  have h₄₂ : (42 * n : ℕ) | (41 * n + 42 : ℕ) := by
    norm_cast at h₄₁ ⊢
    <;> (try omega)
    <;> (try {
      cases' h₄₁ with k hk
      use k
      <;> (try ring_nf at hk ⊢)
      <;> (try norm_cast at hk ⊢)
      <;> (try omega)
    })
    <;> (try {
      simp_all [mul_comm]
      <;> ring_nf at *
      <;> norm_cast at *
      <;> omega
    })
  simpa [mul_comm] using h₄₂

exact h₄

theorem amc12b_2002_p4 (n : ℕ) (h₀ : 0 < n) (h₁ : (1 / 2 + 1 / 3 + 1 / 7 + 1 / ↑n : ℚ).den
↪  = 1) : n = 42 := by
  have h_sum_is_frac : (1 / 2 + 1 / 3 + 1 / 7 + 1 / ↑n : ℚ) = (↑(41 * n + 42)) / ↑(42 * n)
↪   := by
    exact h_sum_is_frac_amc12b_2002_p4 n h₀

  have h_divides : 42 * n | 41 * n + 42 := by
    exact h_divides_amc12b_2002_p4 n h₀ h₁ h_sum_is_frac

  have h_k_is_one : ∀ k : ℕ, 41 * n + 42 = k * (42 * n) → k = 1 := by
    exact h_k_is_one_amc12b_2002_p4 n h₀

  rcases h_divides with ⟨k, hk⟩
  rw [mul_comm (42 * n)] at hk

  have k_one : k = 1 := by
    exact k_one_amc12b_2002_p4 n h₀ k hk h_k_is_one

  rw [k_one, one_mul] at hk
  rw [show 42 * n = 41 * n + n by ring] at hk

  exact (Nat.add_left_cancel hk).symm
```

## A.6 INFERENCE TIME COMPUTE

**PutnamBench.** Our main paper results for HILBERT with Gemini 2.5 Pro were obtained using exhaustive compute with multiple retries per problem, without tracking inference-time compute. To better understand the inference-time compute scaling behavior, we rerun PutnamBench with strictly one attempt per problem for both HILBERT with Gemini 2.5 Pro and `gpt-oss-120b`, obtaining a pass rate of 55.9% and 13.3% respectively. The results are reported in Figure 7.

As observed on MiniF2F, performance improves continuously with additional inference-time compute for both model configurations. However, the more capable Gemini 2.5 Pro is substantially more compute-efficient than `gpt-oss-120b`, achieving higher performance at equivalent compute budgets while also being capable of sustaining more reasoning turns to tackle harder problems. Compared to MiniF2F, the most challenging PutnamBench problems require several orders of magnitude more tokens, reflecting the greater difficulty of the benchmark. Finally, the compute distribution exhibits a pronounced long-tail, with the median number of LLM calls required being far lower than the mean (Table 5), indicating that while most problems are resolved efficiently, a small number of hard problems consume disproportionately large amounts of compute.

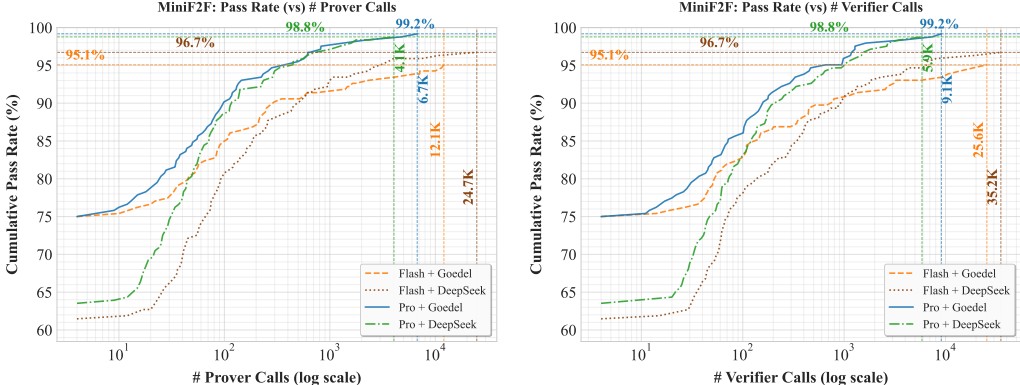

Figure 6: **Pass rate (vs) Prover and Verifier Calls.** We plot the pass-rate for HILBERT on MiniF2F as a function of (left) the number of calls to the Prover (right) the number of calls to the Verifier, per sample.

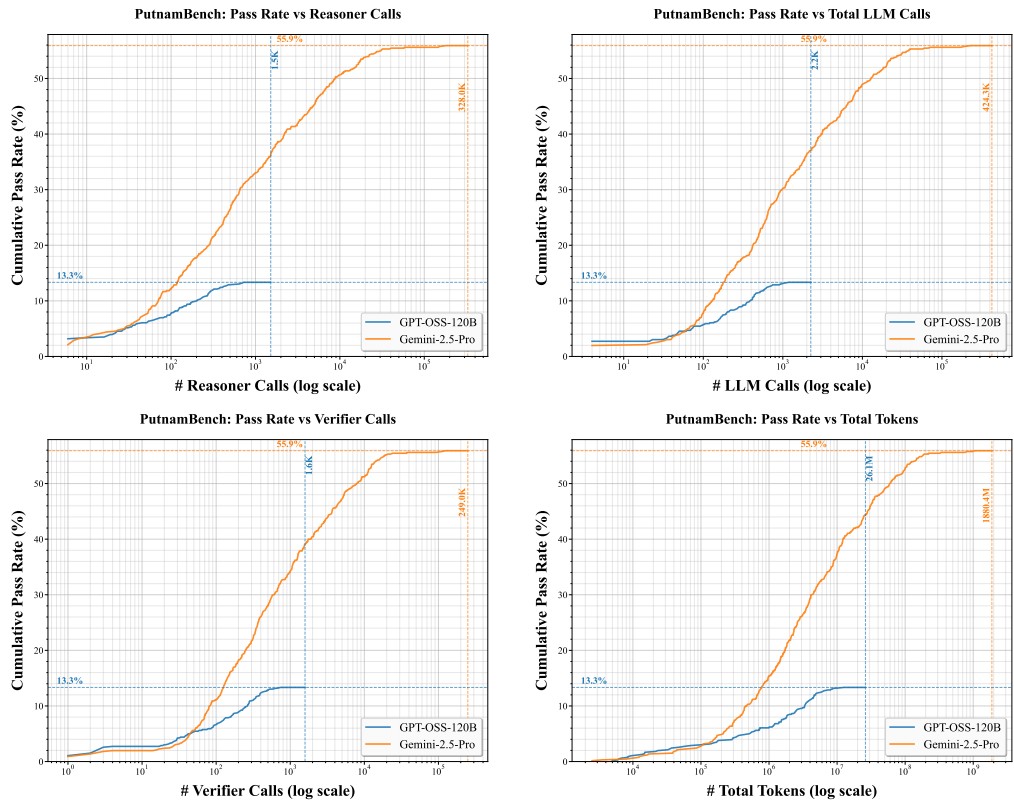

Figure 7: **Pass rate (vs) Inference-time Budget.** We plot the pass-rate for HILBERT on PutnamBench as a function of (top-left) the number of Reasoner calls (top-right) the total number of LLM (Reasoner + Prover) calls per sample (bottom-left) number of verifier calls (bottom-right) total number of tokens.

| Run | Avg #tokens/problem | Avg #verifier calls | Avg #reasoner calls | Avg # LLM calls |
|---|---|---|---|---|
| Gemini-2.5-Flash + Goedel-Prover-V2-32B | 3.3M | 427.5 | 403.9 | 635.0 |
| Gemini-2.5-Flash + DeepSeek-Prover-V2-7B | 2.7M | 475.3 | 360.3 | 646.4 |
| Gemini-2.5-Pro + Goedel-Prover-V2-32B | 0.9M | 146.3 | 133.7 | 230.8 |
| Gemini-2.5-Pro + DeepSeek-Prover-V2-7B | 0.8M | 137.0 | 159.5 | 235.5 |
| GPT-OSS-120B + Goedel-Prover-V2-32B | 0.1M | 15.9 | 7.6 | 22.0 |

Table 4: HILBERT inference-time compute summary statistics on the MiniF2F dataset, computed over successful proofs.

| Run | Avg #tokens/problem | Avg #verifier calls | Avg #reasoner calls | Avg #LLM calls | Median #LLM calls | Median #reasoner calls |
|---|---|---|---|---|---|---|
| Gemini-2.5-Pro + Goedel-Prover-V2-32B | 24.4M | 3019.4 | 4083.5 | 5352.2 | 817.0 | 549.0 |
| GPT-OSS-120B + Goedel-Prover-V2-32B | 2.0M | 156.2 | 129.2 | 235.4 | 160.5 | 67.5 |

Table 5: HILBERT inference-time compute summary statistics on PutnamBench, computed over successful proofs.

| Failure Stage | OpenAI `gpt-oss-120b` | Gemini 2.5 Pro |
|---|---|---|
| Sketch generation | 497 (87.04%) | 79 (27.24%) |
| Sketch assembly | 68 (11.91%) | 81 (27.93%) |
| Shallow solve | 6 (1.05%) | 130 (44.83%) |

Table 6: Failure mode breakdown on PutnamBench.

## A.7 FAILURE MODES ON PUTNAMBENCH

To better understand the failure modes of HILBERT on PutnamBench, we log every LLM call and verifier call during inference to determine the bottleneck to model performance. We then analyze the last step of HILBERT before failure and classify them as either sketch generation, sketch assembly or shallow solve. While sketch generation/assembly failures indicate the model's inability to construct a usable sketch and decomposition strategy, shallow solve errors indicate failure to discharge already-generated subgoals into valid Lean proofs. Table 6 indicates the percentage of number and percentage of problems failed at each stage.

We observe that the failure profile is qualitatively different across models. GPT-OSS-120B failures are overwhelmingly concentrated at sketch generation (497/571, 87.04%) and assembly (68/660, 11.91%), with almost none reaching shallow solve (6/571, 1.05%), suggesting the model rarely produces a usable proof structure in the first place. Gemini, by contrast, fails much later in the pipeline. Most failures occur at shallow solve (130/290, 44.83%), followed by sketch assembly (81/290, 27.93%) and sketch generation (79/290, 27.24%). This pattern indicates that Gemini more reliably constructs a decomposition strategy but frequently struggles to close the final verification loop. These failure modes suggest that allocating more attempts per problem could help mitigate a meaningful fraction of these failures.

## A.8 PROOF LENGTHS

Figures 8 and 9 show the distribution of proof lengths generated by HILBERT on the MiniF2F and PutnamBench datasets, respectively. For comparison, Figure 8 also includes proof lengths from DeepSeek-Prover-V2-671B on MiniF2F problems.

On MiniF2F, HILBERT generates substantially longer proofs than DeepSeek-Prover-V2-671B, with an average length of 247 lines compared to 86.7 lines. Notably, HILBERT produces one proof spanning 8,313 lines, demonstrating its capacity for tackling hard problems.

This trend toward longer proofs is even more pronounced on PutnamBench, where HILBERT achieves an average proof length of 1,454 lines. The longest proof on this dataset exceeds 15,000 lines of code. The ability to consistently generate such extensive proofs likely contributes to HILBERT's superior performance on PutnamBench compared to baseline methods, as longer proofs may reflect more thorough exploration of intermediate steps necessary for a complete Lean proof.

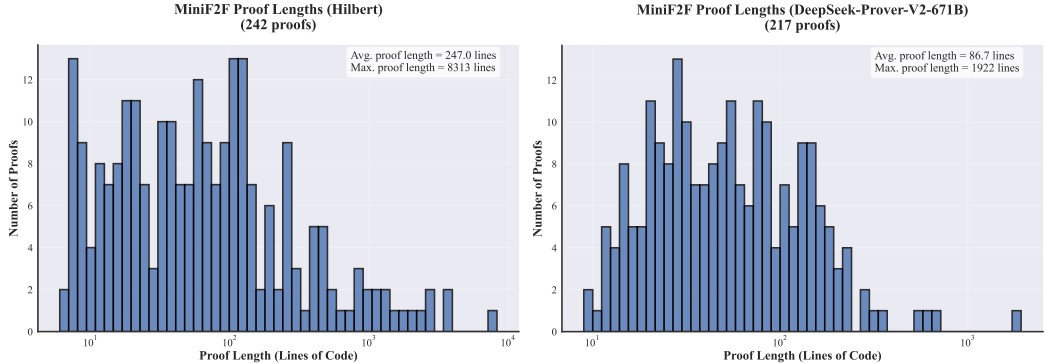

Figure 8: Lengths of proofs generated by (left) HILBERT (Gemini 2.5 Pro + Goedel-Prover-V2) (right) DeepSeek-Prover-V2 671B for problems from MiniF2F.

**PutnamBench Proof Lengths**
**(462 proofs)**

Avg. proof length = 1454.5 lines
Max. proof length = 15306 lines

Figure 9: Lengths of proofs generated by HILBERT (Gemini 2.5 Pro + Goedel-Prover-V2) for problems from PutnamBench.

