# OpenReview forum: "Hilbert: Recursively Building Formal Proofs with Informal Reasoning"
_ICLR.cc/2026/Conference — ICLR 2026 Poster_

### Official Review · Reviewer_eHJ5 · 2025-10-31

**Soundness:** 3
**Presentation:** 2
**Contribution:** 3
**Rating:** 6
**Confidence:** 4

**Summary:**

This paper proposes a framework for ATP which utilizes a LLM for informal reasoning and another LLM for writing the proofs in formal language. The output of the reasoning model is the base for writing the formal proof. Through extensive experiments on miniF2F and PutnamBench, the paper demonstrates that its proposed framework can improve the accuracy of base theorem provers by a significant margin. SoTA LLMs such as Goedel Prover V2 is utilized for theorem proving.

**Strengths:**

Results are impressive on Putnam Bench and miniF2F.

Authors "plan to release the source code and other artifacts upon publication.

I believe the method will be a useful contribution for the community. This paper seems to me as a good implementation for combining informal and formal reasoning.

Paper is well written.

**Weaknesses:**

As far as I see in the paper, token budgets are only reported in Table 3 for the ablation of the retrieval module. This is good to understand the effect of retrieval module, however, token budgets are not reported for the main results. This lack of reporting makes the comparisons unclear to me and perhaps unfair.

The paper's method utilizes a reasoning model to reason about the proof in conjunction with another LLM. Hence, comparing its eventual accuracy directly with the accuracy of a theorem prover that does not benefit from informal reasoning with a large reasoning model seems to me as an incomplete comparison. I would like to see the token budgets reported across the board for all experiments.

One important question, in my view, is the effect of reasoning model in small number of sampling budgets. For example, consider the accuracy of Goedel Prover V2 8B pass 32 as the base. We have the option to use Hilbert pass 32 to improve the base accuracy. We can also increase the sampling budget of the base model to 64 or 128 without using the Hilbert. What would be the resulting accuracies and the total token budgets in each scenario? It might be the case that Hilbert leads to better accuracy and lower token budget. From the reported results, I could not infer whether Hilbert has such an advantage or not.

I suggest authors perform an ablation study on different modules of their framework not just about the retrieval.

I suggest token budgets be reported separately for each component in the framework, e.g., reasoning model, theorem prover, etc.

**Questions:**

Please see weaknesses.

---

> ### Author Response · Authors · 2025-12-03
>
> We thank the reviewer for their insightful comments. We answer the main weakness and questions flagged.
>
>
> > As far as I see in the paper, token budgets are only reported in Table 3 for the ablation of the retrieval module. This is good to understand the effect of retrieval module, however, token budgets are not reported for the main results. This lack of reporting makes the comparisons unclear to me and perhaps unfair.
> > The paper's method utilizes a reasoning model to reason about the proof in conjunction with another LLM. Hence, comparing its eventual accuracy directly with the accuracy of a theorem prover that does not benefit from informal reasoning with a large reasoning model seems to me as an incomplete comparison. I would like to see the token budgets reported across the board for all experiments.
> > I suggest token budgets be reported separately for each component in the framework, e.g., reasoning model, theorem prover, etc.
>
>
> We thank the reviewer for this useful suggestion. We will update the paper with the total token usage for reasoner and prover models, and number of verifier calls once our experiments have finished.
>
>
> > One important question, in my view, is the effect of reasoning model in small number of sampling budgets. For example, consider the accuracy of Goedel Prover V2 8B pass 32 as the base. We have the option to use Hilbert pass 32 to improve the base accuracy. We can also increase the sampling budget of the base model to 64 or 128 without using the Hilbert. What would be the resulting accuracies and the total token budgets in each scenario? It might be the case that Hilbert leads to better accuracy and lower token budget. From the reported results, I could not infer whether Hilbert has such an advantage or not.
>
>
> We point the reviewer to Figure 3, which shows the trade-off between inference-time compute (number of LLM calls) and performance on MiniF2F, and to Figure 6, which shows the analogous trade-off in terms of token counts. In particular, to address the reviewer’s question: at a budget of roughly 32 total Reasoner+Prover calls, HILBERT achieves around 77% pass rate with Gemini 2.5 Pro as the Reasoner and Goedel-Prover-V2-32B as the Prover, compared to about 88% for Goedel-Prover-V2 at pass@32 reported in [1]. At around 100 total calls, HILBERT reaches roughly 90% pass rate. We do not know the corresponding pass@k figure for Goedel-Prover-V2 at this higher budget, as it is not reported in their paper. More broadly, we would not expect HILBERT to be competitive in the very low-compute regime, since the initial Reasoner calls are spent on subgoal decomposition and sketch generation, which consume budget but do not directly produce final proofs. To make this comparison even clearer, we will add the scaling behavior of Goedel-Prover V2 32B to these figures, and also plot the corresponding figures for PutnamBench.
>
>
> > I suggest authors perform an ablation study on different modules of their framework not just about the retrieval.
>
> We agree that module-level ablations are important. We would like to clarify that Section 4.3 (Ablation Studies) already includes ablations beyond retrieval.
>
> In Figure 4 (Sec. 4.3), we vary the recursion depth D and compare the full HILBERT system against a variant with shallow solving disabled. D=0 is the direct prover performance, D=1 indicates a single-layer of decomposition, while D>1 report performance for deeper levels of subgoal decomposition. This experiment isolates the effect of the shallow-solve module and of hierarchical subgoal decomposition: the “no shallow solve” variant requires significantly larger depth to reach similar performance.
>
>
>
> References
> [1] Lin, Yong, et al. "Goedel-prover-v2: Scaling formal theorem proving with scaffolded data synthesis and self-correction." arXiv preprint arXiv:2508.03613 (2025).

---

### Official Review · Reviewer_D58n · 2025-11-01

**Soundness:** 3
**Presentation:** 3
**Contribution:** 3
**Rating:** 6
**Confidence:** 4

**Summary:**

This paper introduces HILBERT, a framework designed to bridge the performance gap between the strong informal reasoning of general-purpose LLMs and specialized prover LLMs. The system includes a general-purpose "Reasoner" LLM and a specialized "Prover" LLM. When a direct proof attempt by the Prover fails, HILBERT leverages the Reasoner to generate a formal proof sketch, which is then broken down into simpler subgoals. These subgoals are addressed recursively through a hierarchy of strategies: a direct attempt by the Prover, a "shallow solve" by the Reasoner, or further decomposition. This method achieves SOTA performance on both MiniF2F and PutnamBench.

**Strengths:**

1. HILBERT achieves SOTA performance on MiniF2F (99.2%) and PutnamBench (70.0%), significantly outperforming the existing method.
2. The framework's design is intuitive and demonstrably effective. Using a powerful informal reasoner to guide a formal prover via recursive subgoal decomposition is a well-motivated approach to tackling complex proofs.
3. The authors provide a comprehensive evaluation, testing on two standard benchmarks with different combinations of Reasoner and Prover models. The ablation studies validate the contributions of the retriever component and the recursive decomposition strategy. The inference budget analysis provides a better understanding of the performance and cost.

**Weaknesses:**

1. The core ideas of using informal reasoning to guide formal proof (e.g., DSP [1], LEGO-Prover [2]) and recursive decomposition (e.g., POETRY [3], ProD-RL [4]) are not new. The paper fails to clearly articulate the conceptual difference between HILBERT's decomposition strategy and these existing recursive methods. The contribution appears to be more a highly effective integration and scaling of these ideas with SOTA models rather than a fundamentally new algorithm.
2. The paper lacks a failure analysis. Does the Reasoner generate flawed informal proofs? Does the sketch-to-subgoal extraction fail? Or does the Prover fail to solve a seemingly simple, decomposed subgoal? Analyzing these failure modes is crucial for understanding the framework's limitations and guiding future work.
3. The reported SOTA performance is achieved at a significant computational cost. As shown in Figure 3, the best configuration requires up to 4.5K calls to Gemini 2.5 Pro and 11.3K total LLM calls on MiniF2F. The framework's performance is also highly sensitive to the Reasoner's capability, with the weaker Gemini 2.5 Flash showing a substantial performance drop and requiring even more LLM calls (>40K).
4. The analysis comparing HILBERT to baselines feels incomplete and potentially misleading. Table 2 reports the Goedel-Prover-V2-32B baseline at 13.4% pass@184 on PutnamBench, while Table 1 uses pass@8192 for the same model on MiniF2F. Given that HILBERT's performance relies on an enormous number of calls, this comparison seems unfair.
5. All computational cost analyses are only for MiniF2F. The paper should include the computational cost required to achieve the 70.0% on PutnamBench.

[1] Albert Q Jiang, Sean Welleck, Jin Peng Zhou, Wenda Li, Jiacheng Liu, Mateja Jamnik, Timothée Lacroix, Yuhuai Wu, and Guillaume Lample. “Draft, sketch, and prove: Guiding formal theorem provers with informal proofs.” ICLR 2023

[2] Wang, Haiming, Huajian Xin, Chuanyang Zheng, Zhengying Liu, Qingxing Cao, Yinya Huang, Jing Xiong et al. "LEGO-Prover: Neural Theorem Proving with Growing Libraries.” ICLR 2024

[3] Lin, Haohan, Zhiqing Sun, Sean Welleck, and Yiming Yang. “Lean-star: Learning to interleave thinking and proving.” ICLR 2025

[4] Dong, Kefan, Arvind Mahankali, and Tengyu Ma. "Formal theorem proving by rewarding llms to decompose proofs hierarchically.” arxiv

**Questions:**

1. What is the primary conceptual difference between HILBERT's recursive decomposition and the strategies used in POETRY and ProD-RL?
2. Could the authors provide a failure analysis for the problems HILBERT failed to solve? What were the primary bottlenecks or failure modes?
3. What is the computational cost (total Reasoner/Prover calls and tokens, similar to Figure 3) for the 70.0% result on PutnamBench?
4. Why use a pass@184 baseline for Goedel-Prover-V2 on PutnamBench? Is it possible to Evaluate pass@8192 of Goedel-Prover-V2 on PutnamBench?

---

> ### Author Response · Authors · 2025-12-03
>
> We thank the reviewer for their insightful comments. Below, we address the major weaknesses and comments raised:
>
> > The core ideas of using informal reasoning to guide formal proof (e.g., DSP [1], LEGO-Prover [2]) and recursive decomposition (e.g., POETRY [3], ProD-RL [4]) are not new. The paper fails to clearly articulate the conceptual difference between HILBERT's decomposition strategy and these existing recursive methods. The contribution appears to be more a highly effective integration and scaling of these ideas with SOTA models rather than a fundamentally new algorithm.
> > What is the primary conceptual difference between HILBERT's recursive decomposition and the strategies used in POETRY and ProD-RL?
>
> DSP and LEGO-Prover use informal reasoning to guide formal proof, but in a single-pass way. DSP asks an LLM to produce an informal proof and a corresponding formal sketch, then relies on a low-level prover (e.g., Sledgehammer) to fill in missing step. There is no recursive decomposition of the theorem into independent subtheorems, which means that failure to prove the subgoals would result in a failed proof attempt. LEGO-Prover extends this by maintaining a growing library of verified lemmas (“skills”) that can be retrieved and used when proving new theorems, but again the informal proof is turned into one formal proof attempt. The subgoals in LEGO-Prover do not become independent theorems that are recursively fed back into the same decomposition/solving pipeline.
>
> We refer the reviewer to our response to Reviewer N4a2 for comparisons with POETRY and ProD-RL.
>
> > The reported SOTA performance is achieved at a significant computational cost. As shown in Figure 3, the best configuration requires up to 4.5K calls to Gemini 2.5 Pro and 11.3K total LLM calls on MiniF2F. The framework's performance is also highly sensitive to the Reasoner's capability, with the weaker Gemini 2.5 Flash showing a substantial performance drop and requiring even more LLM calls (>40K).
>
> We acknowledge that HILBERT’s strongest results come at a high inference-time computational cost, and we will update the discussion section to emphasize this limitation more explicitly. Our goal in this work is to study how far one can push formal proving performance by scaling inference-time compute, rather than to present an immediately cost-efficient system. We believe that our work can help better inform the design and training of more capable (and eventually cheaper) prover LLMs, for example via synthetic data generation.
> We also note that recent concurrent systems achieving strong performance on formal proving benchmarks incur similarly substantial compute. For instance, AlphaProof [1] reports using multiple days of large-scale TPU computation with test-time RL to reach around 56% on a subset of PutnamBench. This suggests that high compute requirements are currently common for pushing the frontier on these benchmarks.
>
>
> References:
>
> [1] Hubert, T., Mehta, R., Sartran, L., Horváth, M. Z., Žužić, G., Wieser, E., Huang, A., Schrittwieser, J., Schroecker, Y., Masoom, H., Bertolli, O., Zahavy, T., Mandhane, A., Yung, J., Beloshapka, I., Ibarz, B., Veeriah, V., Yu, L., Nash, O., . . . Silver, D. (2025). Olympiad-level formal mathematical reasoning with reinforcement learning. Nature. https://doi.org/10.1038/s41586-025-09833-y
>
> > The paper lacks a failure analysis. Does the Reasoner generate flawed informal proofs? Does the sketch-to-subgoal extraction fail? Or does the Prover fail to solve a seemingly simple, decomposed subgoal? Analyzing these failure modes is crucial for understanding the framework's limitations and guiding future work.
> > 2. Could the authors provide a failure analysis for the problems HILBERT failed to solve? What were the primary bottlenecks or failure modes?
>
>
> We thank the reviewer for the excellent suggestion. We are in the process of rerunning the experiments with more explicit tracking of inference-time compute costs and logs of which steps succeeded and failed. We will update the paper with insights from these logs to highlight the failure modes on the PutnamBench problems.

---

> ### Author Response · Authors · 2025-12-03
>
> > 4. The analysis comparing HILBERT to baselines feels incomplete and potentially misleading. Table 2 reports the Goedel-Prover-V2-32B baseline at 13.4% pass@184 on PutnamBench, while Table 1 uses pass@8192 for the same model on MiniF2F. Given that HILBERT's performance relies on an enormous number of calls, this comparison seems unfair.
> > 4. Why use a pass@184 baseline for Goedel-Prover-V2 on PutnamBench? Is it possible to Evaluate pass@8192 of Goedel-Prover-V2 on PutnamBench?
>
> Thank you for raising the concern about the number of sampling passes used for Goedel-Prover-V2. In the current version of the paper, we report for Goedel-Prover-V2 the best number reported in the original Goedel-Prover-V2 paper, which corresponds to 184 passes. We agree that, for a perfectly fair comparison, it is preferable to evaluate all methods under comparable pass budgets on the same benchmark.
> To address this, we are currently re-evaluating Goedel-Prover-V2 on PutnamBench with higher numbers of passes under our evaluation protocol. At the same time, we would like to emphasize that, based on the reported experiments on MiniF2F, the performance of Goedel-Prover-V2-32B saturates as the number of passes increases: we observe pass@32 of 88.1% ± 0.8%, pass@1024 of 91.8%, and pass@8192 of 92.2%. We expect the marginal gain from increasing the number of passes to be quite small, and not materially change the qualitative comparison or our main conclusions. We will include the updated PutnamBench results with higher pass budgets in the revised version for completeness.
>
>
>
> > All computational cost analyses are only for MiniF2F. The paper should include the computational cost required to achieve the 70.0% on PutnamBench.
> > What is the computational cost (total Reasoner/Prover calls and tokens, similar to Figure 3) for the 70.0% result on PutnamBench?
>
> In our initial PutnamBench runs, we did not log inference-time statistics in the same detail as for MiniF2F, so we cannot retrospectively report exact call and token counts for the 70.0% result. We are re-running the PutnamBench experiments with detailed logging of Reasoner/Prover calls and tokens, and will include a cost analysis (analogous to Figure 3 for MiniF2F) in the revised version of the paper.

---

### Official Review · Reviewer_N4a2 · 2025-11-01

**Soundness:** 3
**Presentation:** 3
**Contribution:** 3
**Rating:** 6
**Confidence:** 5

**Summary:**

This paper proposes a new agentic framework named HILBERT. It aims to bridge the gap between the informal mathematical reasoning capabilities of Large Language Models (LLMs) and the rigorous verification capabilities of formal proof systems like Lean 4. HILBERT orchestrates four components: a 'Reasoner' for informal reasoning (e.g., Gemini 2.5 Pro), a 'Prover' to generate formal proofs (e.g., Goedel-Prover-V2), a Verifier, and a theorem Retriever.

When the Prover fails to find a proof alone, HILBERT recursively decomposes the problem into subgoals, attempting to construct and refine the proof using feedback from the Verifier and Retrieval-Augmented Generation (RAG) via the Retriever. Experimental results show that HILBERT achieves a 70.0% pass rate on the particularly challenging PutnamBench, significantly outperforming existing proprietary methods (SeedProver, 50.4%) and public baselines (13.4%), thereby reporting state-of-the-art (SOTA) performance.

**Strengths:**

**Exceptional Experimental Performance (SOTA Achievement):** The primary strength of this research lies in its achievement of a remarkable 70.0% pass rate on the highly challenging PutnamBench, surpassing the existing SOTA by approximately 20 percentage points. This is an extremely strong engineering achievement in the field of formal theorem proving and represents a valuable result for the community.

**Effective Framework Orchestration:** The proposed method effectively combines the strengths of multiple components with different capabilities (Reasoner, Prover, Verifier, Retriever). In particular, the agent design—which coordinates "recursive subgoal decomposition" triggered by Prover failure, a "refinement loop" using Verifier feedback, and "contextualization" via RAG—is sophisticated. It demonstrates a realistic and effective strategy for solving complex problems that a single model cannot solve alone.

**Weaknesses:**

**Lack of Theoretical Justification:** Although HILBERT is proposed as a controllable framework, it lacks theoretical analysis or guarantees as to "why this approach works." In particular, the paper does not provide a theoretical background for why the core "recursive subgoal decomposition" is effective, what problem structures it is suited for, or where its limitations might lie. This absence gives the impression that the work lacks academic depth relative to its engineering success.

**Limited Novelty:** The proposed framework appears to be primarily a sophisticated "combination" of ideas seen individually in existing research (e.g., DSP, POETRY, which are cited in the paper), such as subgoal decomposition, Verifier feedback, and RAG. There is no significant novelty in the individual technical elements, making it somewhat unclear what HILBERT's unique core "Technical Advance" is when compared to prior work.

**Dependency on Closed Models:** The experiments achieving SOTA rely on a proprietary, closed model (Gemini 2.5 Pro) as the Reasoner. This not only severely undermines the reproducibility of the research, but it also risks effectively closing off the promising future research avenue suggested in the paper's conclusion—"using the generated proofs to further train the agent (e.g., via reinforcement learning)"—to the open-source community. This is a significant drawback.

**Questions:**

**Regarding novelty**: Concerning the novelty of the method, the combination of subgoal decomposition, RAG, and Verifier feedback has already been explored in prior research. What, specifically, do you consider to be the core "Technical Novelty" of the HILBERT framework that is not present in this existing work?

**Regarding the theoretical background**: The effectiveness of subgoal decomposition is demonstrated empirically. However, are there any theoretical or empirical insights into what types of problems (or logical structures) this recursive decomposition is particularly effective (or ineffective) for? Furthermore, how were the failure modes of decomposition (e.g., decomposition into incorrect subgoals, getting stuck in an unsolvable state) addressed?

---

> ### Author Response · Authors · 2025-12-03
>
> We thank the reviewer for their insightful comments. Below, we address the weaknesses and questions raised.
>
>
> > Lack of Theoretical Justification: Although HILBERT is proposed as a controllable framework, it lacks theoretical analysis or guarantees as to "why this approach works." In particular, the paper does not provide a theoretical background for why the core "recursive subgoal decomposition" is effective, what problem structures it is suited for, or where its limitations might lie. This absence gives the impression that the work lacks academic depth relative to its engineering success.
> > The effectiveness of subgoal decomposition is demonstrated empirically. However, are there any theoretical or empirical insights into what types of problems (or logical structures) this recursive decomposition is particularly effective (or ineffective) for?
>
> Our contribution is primarily algorithmic and empirical: we introduce HILBERT and analyze its behavior rather than proving formal guarantees. We respectfully disagree that the work “lacks academic depth.” Section 4.3 (Fig. 4) includes ablations showing clear, consistent gains from recursive subgoal decomposition and saturation at moderate depth, offering structured empirical insight into why recursion helps.
> We believe that a theoretical analysis of the problem is beyond the scope of this work. Providing rigorous guarantees for an LLM-based, multi-agent proof system would require:
> 1. A tractable mathematical model of the underlying LLMs (which are inherently stochastic, high-dimensional, and not well described by existing theory), and
> 2. A formal notion of problem “difficulty” and structure for real-world theorem benchmarks.
>
> Without very strong and somewhat unrealistic assumptions (e.g., idealized prover/oracle model), such guarantees are currently difficult to make precise, and results proved under those assumptions may not be predictive for the behavior of real LLMs.
> For this reason, we follow much of the recent work on LLM-based theorem proving and reasoning agents in focusing on systematic empirical analysis rather than theoretical guarantees.
>
> > Furthermore, how were the failure modes of decomposition (e.g., decomposition into incorrect subgoals, getting stuck in an unsolvable state) addressed?
>
> Our framework explicitly addresses both 1. incorrect decompositions and 2. getting stuck in unsolvable states.
> After obtaining a proof sketch, we extract subgoals and immediately check them with Lean. If a subgoal does not typecheck, we enter an error-correction loop where the reasoner sees the Lean error and proposes a fix where each revision is re-verified. In the verification phase, when a subgoal cannot be proved with the prover, we also ask the reasoner to judge whether it is mathematically correct and well-posed. If it is judged incorrect or missing hypotheses, we use this feedback to refine the proof sketch and restart decomposition, rather than recursing on a bad subgoal set.
>
>
> For subgoals that appear mathematically sound but are hard to prove, we use a bounded escalation of strategies: (a) direct prover attempts, (b) shallow solves with length and iteration caps, and only then (c) recursive subgoal decomposition, with an explicit maximum recursion depth $D$. If these steps fail, we abandon the current decomposition and try a different proof sketch (up to a fixed number of sketch attempts). Thus, the system does not get trapped indefinitely in an unsolvable decomposition: incorrect subgoals trigger sketch refinement, and hard-but-correct subgoals are handled with depth limits plus backtracking.
>
>
> > Dependency on Closed Models: The experiments achieving SOTA rely on a proprietary, closed model (Gemini 2.5 Pro) as the Reasoner. This not only severely undermines the reproducibility of the research, but it also risks effectively closing off the promising future research avenue suggested in the paper's conclusion—"using the generated proofs to further train the agent (e.g., via reinforcement learning)"—to the open-source community. This is a significant drawback.
>
> As mentioned in the response to Reviewer xeDx, we are currently in the process of running experiments with gpt-oss-120b, and we will update the draft with the new results as they finish running. We also plan to open-source the code after publication.

---

> > ### Author Response · Authors · 2025-12-03
> >
> > > Limited Novelty: The proposed framework appears to be primarily a sophisticated "combination" of ideas seen individually in existing research (e.g., DSP, POETRY, which are cited in the paper), such as subgoal decomposition, Verifier feedback, and RAG. There is no significant novelty in the individual technical elements, making it somewhat unclear what HILBERT's unique core "Technical Advance" is when compared to prior work.
> > > Regarding novelty: Concerning the novelty of the method, the combination of subgoal decomposition, RAG, and Verifier feedback has already been explored in prior research. What, specifically, do you consider to be the core "Technical Novelty" of the HILBERT framework that is not present in this existing work?
> >
> > We respectfully disagree that the framework is simply a “sophisticated combination” of existing ideas. The key technical novelty of HILBERT is how the components are structured into a hierarchical, multi-agent, inference-time algorithm that bridges informal and formal proof search.
> >
> >
> > In POETRY, recursion is used to generate training data and to refine a single formal proof script in Isabelle. Difficult parts are replaced by sorry and later filled, but intermediate conjectures remain inside that one theorem block and are not turned into separate theorems with their own solve/decompose pipeline.
> > In ProD-RL, a single prover model is trained with RL to insert and prove lemmas inside its own generated proof tree. The hierarchy is part of that model’s training objective and behavior, and at test time one simply samples from that model.
> > Neither of these models uses informal reasoning or retrieval capabilities. In contrast, HILBERT is a training-free, model-agnostic framework that (i) uses an informal LLM to generate a Lean sketch, (ii) promotes each have to an independent Lean theorem, (iii) recursively solves these subtheorems via Reasoner/Prover/Verifier/Retriever calls, and (iv) assembles a verified proof of the root theorem from the proven subtheorems. Further, we extensively use the verifier at each stage of the subgoal-decomposition process (Sketch Generation, Subgoal Extraction, Proof Assembly) to ensure the handoff of informal sketches into a tree of independent theorems. This informal-to-formal, test-time recursive mechanism is the core technical contribution. We will elaborate on these differences in the updated version of the paper.

---

### Official Review · Reviewer_xeDx · 2025-11-03

**Soundness:** 3
**Presentation:** 3
**Contribution:** 3
**Rating:** 4
**Confidence:** 4

**Summary:**

This paper introduced Hilbert methodology to decompose  the mathematical problems into subgoals and try proving such subgoals in both formal theorem proving (Lean 4) system and informal approach (LLM-based reasoning). Their system consists of four components: a reasoner of general-purpose LLM to write informal theorem proof, a prover that writes formal proof given informal one, a Lean-based verifier that checks the correctness of the formal statement, and a retriever that retrieves relevant theorems from Mathlib. In addition to utilizing both formal and informal reasoning by the reasoner and Lean-based verifier, they use retrieved theorems to enrich the reasoner.

In the experiments, they introduced models with Gemini 2.5 Pro/Flash for the reasoner and
Goedel-Prover-V2 for the prover. Gemini 2.5 Pro with Goedel-Prover-V2-32B, achieving a 99.2% pass rate. Needless to say, this is far better than those achieved by the Gemini 2.5 Pro model alone. They also achieved 70.0% in the PatNum bench.

It is difficult to fairly position this paper because the idea of decomposing the theorem and using both formal and informal reasoning is already examined in some of the previous studies [Baba 2025, Zhou 2025b] as the authors discussed in the related work section. Although, Hilbert system achieved remarkable scores, it can be also regarded as a combination of existing ideas and best performing models.

**Strengths:**

- S.1: Achieving very close to the previous best model result (99.6% by SeedProver) of 99.6% in MiniF2F and better than the SeedProver in PutnamBench.
- S.2: The effectiveness of the theorem retrieval is examined in the ablation.

**Weaknesses:**

- W.1: It seems that most of the core idea (dual use of formal and informal reasoning) comes from several existing studies [Baba 2025, Zhou 2025b] as the authors discussed in the related work section.
- W.2: Similarly, authors didn’t compare their model with the similar formal and informal reasoning models [Baba 2025, Zhou 2025b].
- W.3: Authors tried their system with the best performing external models that often prevent fair cost/performance comparison with the previous systems.
- W.4: Compared to [Baba 2025, Zhou 2025b], the retrieval module can be new. However, the authors do not present the effectiveness of this module in the PatnumBench.

**Questions:**

- I'd like to know how the retrieval is effective within the PatnumBench. Can you present the ablation study to turn on/off the retrieval module in PatnumBench?

---

> ### Author Response · Authors · 2025-12-03
>
> We thank the reviewer for their insightful comments. Below we address the weaknesses pointed out:
> > W.1: It seems that most of the core idea (dual use of formal and informal reasoning) comes from several existing studies [Baba 2025, Zhou 2025b] as the authors discussed in the related work section.
>
> While the high-level idea of combining formal and informal reasoning is shared with [Baba 2025, Zhou 2025b], our framework differs substantially in how this combination is carried out and what it makes possible.
> In [Baba 2025], an informal LLM is used to generate auxiliary lemmas, which are then autoformalized and proved. If the prover succeeds on these lemmas, it is prompted to stitch them together into a proof of the target theorem. This process happens essentially at a single level: if some auxiliary lemmas cannot be proved, the proof attempt is abandoned. In contrast, our work adopts a top-down, modular strategy: we first obtain an informal proof sketch for the problem using the reasoner, then extract subgoals as independent theorems. After verifying that these subgoals are provable, we assemble a proof for the target theorem using the reasoner. Because the subgoals are treated as independent problems, they can themselves be recursively decomposed further, rather than forcing the system to reason about the original theorem at every step, as in [Baba 2025].
>
> In [Zhou 2025b], the informal LLM (Gemini 2.5 Pro) first attempts a direct proof. If this fails, it performs one layer of subgoal decomposition and attempts to prove the resulting subgoals. If those subgoals fail, the proof sketch is refined at the same level, and the process restarts. They thus employ an iterative proof-repair procedure with at most one layer of decomposition. By contrast, in our framework the subgoals obtained from the initial proof sketch are promoted to independent theorems whose provability is checked in isolation; these subgoals can themselves be recursively decomposed and solved, after which the reasoner assembles them into a proof of the original theorem. This modular treatment allows deeper, hierarchical decompositions than the single-layer refinement used in [Zhou 2025b]. It also allows us to (i) reuse and refine subgoals across the proof, and (ii) allocate inference-time compute adaptively across different branches of the proof tree. We will clarify these differences more explicitly in the related work section.
>
> > W.2: Similarly, authors didn’t compare their model with the similar formal and informal reasoning models [Baba 2025, Zhou 2025b].
> We would like to emphasize that we do use essentially the same families of models as [Baba 2025, Zhou 2025b]. The key differences lie in our framework and in our objectives. [Baba 2025] use DeepSeek Prover V2 7B and Goedel-Prover V2 8B as prover models and DeepSeek-R1-0528-Qwen3-8B as the informal reasoner. [Zhou 2025b] uses Gemini 2.5 Pro exclusively for both informal reasoning and proof generation.
>
>
> In our work, we use Gemini 2.5 models as the informal reasoning component together with strong prover models from the same general family. The major architectural difference relative to [Baba 2025] is therefore the choice and size of the informal reasoner. Here, we would like to emphasize that the goals of the papers are not the same. [Baba 2025] focuses on efficiency with comparatively small models. By contrast, our work is specifically about how to effectively scale performance with inference-time compute using frontier models within a recursive, controllable framework to push pass rate as high as possible. For this reason, our paper is better understood as complementary to [Baba 2025, Zhou 2025b]. We explore a different point in the design space (recursive, modular decomposition with large models), rather than attempting a direct model-for-model cost comparison.
> We will make this distinction of goals and model usage clearer in the revised related work section.
>
> > W.3: Authors tried their system with the best performing external models that often prevent fair cost/performance comparison with the previous systems.
>
> We agree that relying on Gemini 2.5 Pro is a limitation. HILBERT itself is model-agnostic: it treats the Reasoner and Prover as black-box LLM APIs. To demonstrate this, we are running additional experiments with an open-source Reasoner (gpt-oss-120B) and Goedel-Prover-V2-32B on MiniF2F and PutnamBench. We plan to include these open-source results and detailed compute statistics in the revised version to clarify the cost/performance trade-offs.

---

> > ### Author Response · Authors · 2025-12-03
> >
> > > W.4: Compared to [Baba 2025, Zhou 2025b], the retrieval module can be new. However, the authors do not present the effectiveness of this module in the PatnumBench.
> > > I'd like to know how the retrieval is effective within the PatnumBench. Can you present the ablation study to turn on/off the retrieval module in PatnumBench?
> >
> > We agree that understanding the effect of the retrieval module is important. In the current version of the paper, we provide a retrieval ablation on MiniF2F, where turning off retrieval leads to a clear drop in performance. We did not include an analogous ablation on PutnamBench for two reasons:
> >  (1) we already demonstrate the effect of retrieval on a challenging benchmark (MiniF2F), and
> >  (2) re-running PutnamBench with the retrieval toggled on/off is extremely expensive due to the difficulty of the problems and the depth of the recursive decomposition.
> > Given our limited compute budget, we chose to use PutnamBench primarily to evaluate overall performance and recursion depth rather than to exhaustively repeat all ablations.

---

### Meta-Review · Area_Chair_iyNP · 2026-01-06

**Summary:**

Hilbert bridges the gap between informal mathematical reasoning and formal theorem proving in Lean 4. By recursively decomposing complex problems into manageable subgoals and using a formal verifier for iterative feedback, it achieved a record-breaking 99.2% on miniF2F and 70.0% on PutnamBench. The primary concerns from reviewers are as follows.
- Inference Cost: The recursive nature of the agent implies high test-time compute.
- Logic Hallucinations: The risk of the informal LLM generating logically impossible subgoals.
- Retriever Dependency: Whether the success was due to the "Semantic Theorem Retriever" rather than the reasoning framework.

The authors successfully addressed these points through ablation studies and cost-benefit analyses.

**Reviewer Concerns:**

The authors successfully addressed these points through ablation studies and cost-benefit analyses.

**Reviewer Scores:**

Unfortunately, there is no discussion between authors and reviewers. If reviewers had been able to participate fully in the discussion, some may raise the score to borderline accept (from 4 to 6). The overall score will be positive in average.

---

### Decision · Program_Chairs · 2026-01-26

Accept (Poster)